# Optogenetic strategies for high-efficiency all-optical interrogation using blue-light-sensitive opsins

Angelo Forli[1†], Matteo Pisoni[1,2†], Yoav Printz[3†], Ofer Yizhar[3]*, Tommaso Fellin[1]*

[1]Optical Approaches to Brain Function Laboratory, Istituto Italiano di Tecnologia, Genova, Italy; [2]Università di Genova, Genova, Italy; [3]Department of Neurobiology, Weizmann Institute of Science, Rehovot, Israel

**Abstract** All-optical methods for imaging and manipulating brain networks with high spatial resolution are fundamental to study how neuronal ensembles drive behavior. Stimulation of neuronal ensembles using two-photon holographic techniques requires high-sensitivity actuators to avoid photodamage and heating. Moreover, two-photon-excitable opsins should be insensitive to light at wavelengths used for imaging. To achieve this goal, we developed a novel soma-targeted variant of the large-conductance blue-light-sensitive opsin CoChR (stCoChR). In the mouse cortex in vivo, we combined holographic two-photon stimulation of stCoChR with an amplified laser tuned at the opsin absorption peak and two-photon imaging of the red-shifted indicator jRCaMP1a. Compared to previously characterized blue-light-sensitive soma-targeted opsins in vivo, stCoChR allowed neuronal stimulation with more than 10-fold lower average power and no spectral crosstalk. The combination of stCoChR, tuned amplified laser stimulation, and red-shifted functional indicators promises to be a powerful tool for large-scale interrogation of neural networks in the intact brain.

*For correspondence:
ofer.yizhar@weizmann.ac.il (OY);
tommaso.fellin@iit.it (TF)

†These authors contributed equally to this work

Competing interests: The authors declare that no competing interests exist.

## Introduction

During information processing, storage, and retrieval, the electrical activity of most brain circuits is organized in complex spatial and temporal patterns (*Ni et al., 2018*; *Salinas and Sejnowski, 2001*; *Shahidi et al., 2019*; *Shamir and Sompolinsky, 2006*). The coding rules intrinsic to these activity patterns are believed to underlie fundamental operating principles of brain networks (*Averbeck et al., 2006*; *Panzeri et al., 2017*; *Yuste, 2015*). Recent techniques for all-optical interrogation of neural circuits allow for the first time to causally investigate these coding rules with unprecedented spatial resolution in living mice (*Carrillo-Reid et al., 2019*; *Chettih and Harvey, 2019*; *Gill et al., 2020*; *Marshel et al., 2019*; ). In particular, two-photon calcium imaging combined with two-photon optogenetics makes it possible to simultaneously monitor and manipulate dozens of neurons with near single-cell resolution (*Bovetti and Fellin, 2015*; *Chen et al., 2019*; *Forli et al., 2018*; *Mardinly et al., 2018*; *Packer et al., 2015*; *Rickgauer et al., 2014*; *Yang et al., 2018*). However, significant challenges still remain to be addressed to accurately interrogate neural circuits in vivo. First, since the average power of illumination used to stimulate neurons is proportional to the probability of introducing thermal damage to brain tissue and unwanted modulation of neuronal physiology (*Owen et al., 2019*; *Picot et al., 2018*), maximizing the efficiency of optogenetic stimulation by increasing opsin excitation and photocurrent is of utmost importance. Thus, there is a need for the development of novel high-efficiency, large-conductance opsins and efficient illumination strategies for their two-photon activation. Second, the crosstalk between imaging and stimulation, that is, the unwanted activation of opsin-expressing neurons by imaging light, needs to be minimized (*Forli et al., 2018*; *Packer et al., 2015*). This effect is due to the overlap between the two-

photon absorption spectra of the opsin and of the calcium sensor (*Venkatachalam and Cohen, 2014*), and it is typically pronounced for red-shifted opsins, such as C1V1 (*Yizhar et al., 2011*) and ReaChR (*Lin et al., 2013*), which are used in combination with blue-light-sensitive indicators (e.g. GCaMP6 [*Chen et al., 2013*]). This process may lead to non-negligible neuronal activation during two-photon imaging of the blue-light-sensitive indicator (*Packer et al., 2015*). The non-selective excitation of neurons in the field-of-view (FOV) due to this type of crosstalk could promote the formation of unwanted cell-assemblies (*Carrillo-Reid et al., 2016*) and bias the activity in the monitored area, complicating the interpretation of biological results (*Emiliani et al., 2015*). The use of blue-shifted opsins (e.g, ChR2 [*Nagel et al., 2003*] and GtACR2 [*Govorunova et al., 2015*]) combined with red-shifted calcium sensors (e.g. jRCaMP1a [*Dana et al., 2016*]) was proved to be a valid approach, which largely eliminates the effect of the imaging beam on opsin excitation while allowing simultaneous imaging and bidirectional manipulation of neural activity in vivo (*Forli et al., 2018*). However, the limited photocurrent of ChR2 (*Nagel et al., 2003*) prevented the efficient stimulation of multiple neurons with low average power delivery to the brain tissue.

To address this limitation, we describe here a new high-efficiency blue-light-sensitive opsin (stCoChR), derived from the large-conductance opsin CoChR (*Klapoetke et al., 2014*), which was targeted to the soma using the Kv2.1 targeting sequence (*Lim et al., 2000*). We combined stCoChR with holographic two-photon stimulation, low repetition rate laser source excitation tuned at the peak of the opsin absorption spectrum, and imaging of the red-shifted functional indicator jRCaMP1a (*Dana et al., 2016*) in vivo.

## Results

### stCoChR: a high-efficiency, soma-targeted, blue-light-sensitive opsin

In order to achieve efficient optogenetic excitation using minimal light power and to minimize crosstalk between optogenetic stimulation and jRCaMP1a imaging, we used the high-conductance, blue-light-sensitive opsin CoChR (*Klapoetke et al., 2014*). To improve the spatial specificity of stimulation, we restricted the presence of CoChR molecules to the somatic region, such that stimulating one cell body will lead to minimal photocurrents in nearby neuronal projections of other cells. For this purpose, we incorporated a soma-targeting sequence taken from the voltage-gated $K^+$ channel Kv2.1, which was previously found to reduce the presence of channels, including channelrhodopsins, in distal dendrites and axons (*Baker et al., 2016*; *Lim et al., 2000*). To generate soma-targeted CoChR (stCoChR), we appended the Kv2.1 motif C-terminally to the CoChR coding sequence, followed by a fluorophore (mScarlet or eGFP) which was separated from CoChR-Kv2.1 using the self-cleaving peptide P2A (*Kim et al., 2011*), leading to fluorescence labeling of the entire cell (EF1α-CoChR-Kv2.1-P2A-XFP; see Materials and methods). We expressed the non-targeted CoChR (hereafter CoChR) and the soma-targeted stCoChR in sparse sets of cortical neurons of mice using AAV injections (*Figure 1—figure supplement 1*). We also expressed the previously published somatic variant soCoChR (*Shemesh et al., 2017*), in which somatic targeting was achieved using a sequence from the kainate receptor KA2 subunit. We performed whole-cell recordings from opsin-expressing neurons in acute brain slices in order to measure the photocurrents elicited by each variant and their restriction to the somatic region. Upon two-photon (2P) scanning (λ = 940 nm) with a spiral pattern over the soma, photocurrents elicited by stCoChR did not differ from those elicited by CoChR, but were ~130-fold higher than those elicited by soCoChR under the same conditions (*Figure 1A,B* and *Supplementary file 1*- Supplementary Table 1; stCoChR: 1927.4 ± 283.3 pA, n = 10 cells; CoChR: 712.3 ± 188.9 pA, n = 11 cells; soCoChR: 14.9 ± 4.2 pA, n = 10 cells; Kruskal-Wallis test with post hoc comparisons: stCoChR *vs* CoChR, p=0.11; stCoChR *vs* soCoChR, p=3.8E–6). Under one-photon (1P) full-field illumination covering the soma and neurites (illumination area ~0.66 mm$^2$), photocurrents elicited by stCoChR were similar to those of CoChR, but were approximately 40-fold higher than photocurrents elicited by soCoChR (*Figure 1B* and *Supplementary file 1*- Supplementary Table 2; stCoChR: 3709.5 ± 381.2 pA, n = 14 cells; CoChR: 3296.3 ± 452.4 pA, n = 14 cells; soCoChR: 91.8 ± 9.7 pA, n = 11 cells; Kruskal-Wallis test with post hoc comparisons: stCoChR *vs* CoChR, p=0.87; stCoChR *vs* soCoChR, p=2.2E–5). As a measure for restriction of the opsin to the somatic region, we calculated the ratio between the photocurrents observed under 2P soma scanning and under 1P full-field illumination per cell (*Figure 1C* and *Supplementary file 1*-

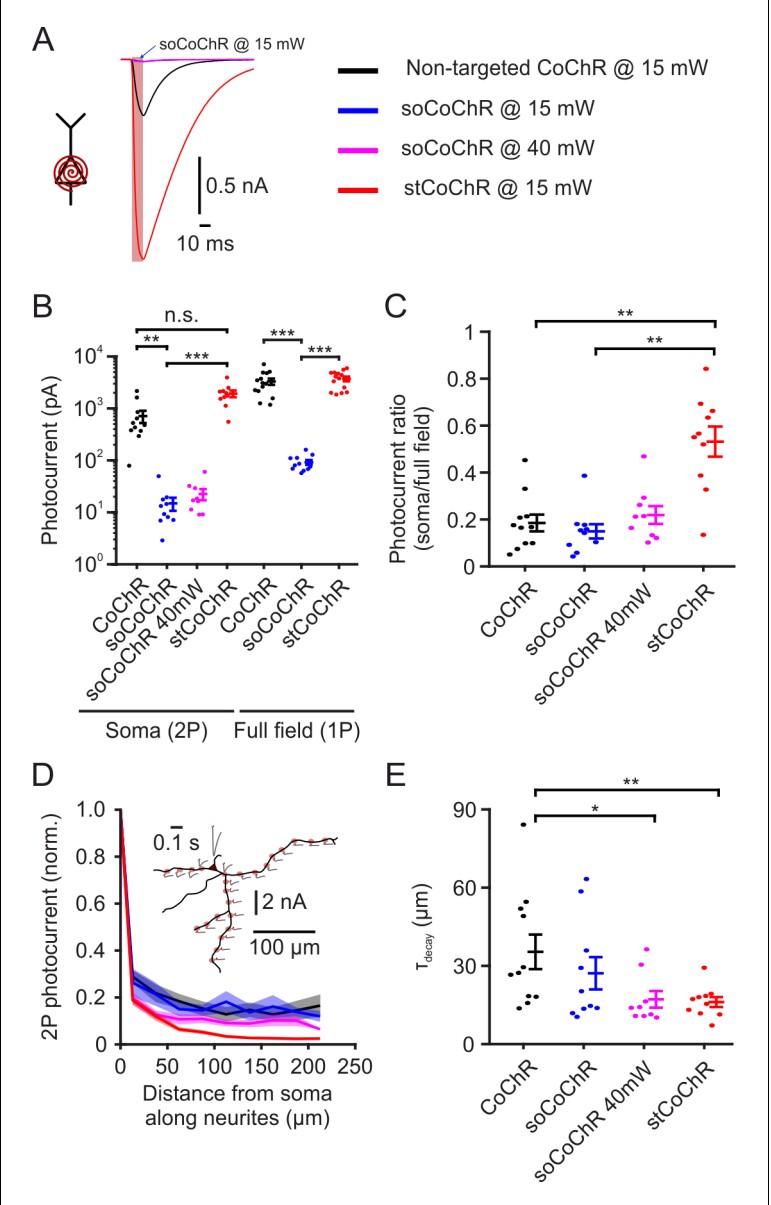

**Figure 1.** Characterization of soma-targeted CoChR variants. (A) Example photocurrents recorded in the acute slice preparation from cortical neurons expressing either non-targeted CoChR, stCoChR or soCoChR upon two-photon scanning with a spiral pattern (10 µm diameter) over the soma (λ = 940 nm, 15 or 40 mW on cell, 10.05 ms scan duration, 80 MHz laser repetition rate). Red shaded background indicates scan period. Traces are from cells displaying the median photocurrent per construct. (B) Photocurrents for each CoChR variant under two-photon spiral scan over the soma (as in A) and under one-photon full-field illumination (470 nm, 0.27 mW/mm$^2$, 500 ms). Please note the logarithmic photocurrent scale. Points are individual cells and vertical lines are mean ± SEM. In this as well as in other figures, *p<0.05; **p<0.01; ***p<0.001; n.s., non-significant. (C) Ratio between the photocurrent under soma spiral stimulation and full-field stimulation per cell, based on the data in B. (D) Photocurrents obtained during two-photon spiral stimulation of selected points along the neurites of cells in the acute slice preparation expressing either variant, normalized to the current obtained at the soma. Distances are measured from the soma along the path of the neurite to the stimulation point and are binned with 25 µm intervals. Shaded background represents SEM. Inset shows a morphological reconstruction of an example cell (black) expressing stCoChR with the stimulation points (shaded red circles) and the absolute photocurrent obtained at each point (gr with red tick indicating stimulation period). (E) Decay length constants of the photocurrent with distance from soma, based on individual cell data from D. p values in B,C,E are based on Kruskal-Wallis test with post hoc comparisons using Tukey's post hoc HSD test.

*Figure 1 continued on next page*

*Figure 1 continued*

The online version of this article includes the following source data and figure supplement(s) for figure 1:

**Source data 1.** Source data for *Figure 1* and *Figure 1—figure supplement 2*.

**Figure supplement 1.** Expression of CoChR and stCoChR.

**Figure supplement 2.** Two-photon photocurrents in neurons expressing different CoChR variants obtained during spiral-scan stimulation of selected points along the neurites.

Supplementary Table 3). We found that this ratio was higher for stCoChR than both CoChR and soCoChR, whereas it did not differ between soCoChR and CoChR (stCoChR: 0.53 ± 0.06, n = 10 cells; CoChR: 0.18 ± 0.04, n = 11 cells; soCoChR: 0.15 ± 0.03, n = 10 cells; Kruskal-Wallis test with post hoc comparisons: stCoChR *vs* CoChR, p=8.4E-3; stCoChR *vs* soCoChR, p=1.3E-3; soCoChR *vs* CoChR, p=0.80), suggesting that the Kv2.1 motif in stCoChR concentrates the opsin molecules at the somatic region. This effect persisted when using a higher light power for 2P stimulation of soCoChR (40 mW instead of 15 mW; *Figure 1B,C*, magenta and *Supplementary file 1*- Supplementary Table 1,3). As another measure for soma restriction, we measured the photocurrents elicited by 2P spiral stimulation of selected points along the neurites of recorded cells (*Figure 1D* and *Figure 1—figure supplement 2*). We found that the photocurrent in cells expressing stCoChR decayed at shorter distances when the stimulation spiral moved away from the soma along the path of the neurites compared with cells expressing CoChR, whereas the decay length constant did not differ between stCoChR and soCoChR and between soCoChR and CoChR (*Figure 1E* and *Supplementary file 1*- Supplementary Table 4; stCoChR: $\tau_{decay}$ = 16.1 ± 1.9 µm, n = 10 cells; CoChR: $\tau_{decay}$ = 35.4 ± 6.6 µm, n = 11 cells; soCoChR: $\tau_{decay}$ = 27.2 ± 6.2, n = 10 cells; Kruskal-Wallis test with post hoc comparisons: stCoChR *vs* CoChR, p=3.8E-2; stCoChR *vs* soCoChR, p=0.46; soCoChR *vs* CoChR, p=0.43). Increasing the power of soCoChR stimulation from 15 mW to 40 mW resulted in a decay length constant similar to that of stCoChR (*Figure 1D,E* and *Supplementary file 1*- Supplementary Table 4; $\tau_{decay}$ = 17.2 ± 3.2, n = 9 cells; Kruskal-Wallis test with post hoc comparisons: soCoChR under 40 mW *vs* CoChR under 15 mW, p=0.02; soCoChR under 40 mW *vs* stCoChR under 15 mW, p=0.94). These results suggest that stCoChR maintains the overall superior photocurrents characteristic of CoChR while a larger fraction of the current arises from opsin molecules in the somatic membrane.

Finally, to improve image-based identification of cells in vivo, we generated an additional variant of stCoChR (CoChR-Kv2.1-P2A-NLS-eGFP) in which the fluorophore was restricted to the nucleus using a nuclear localization signal (NLS) derived from SV 40 large T antigen (*Kalderon et al., 1984*).

## High-efficiency two-photon holographic stimulation of stCoChR-expressing neurons in vivo

We investigated whether stCoChR could be used for efficient two-photon stimulation of neurons in the intact mouse brain in vivo. We first expressed stCoChR in layer 2/3 (L2/3) pyramidal neurons of the mouse cortex (*Figure 2A*) using AAV injections. To stimulate neurons, we incorporated a liquid crystal spatial light modulator (SLM)-based holographic module in the beam path of a two-photon microscope (*Figure 2B*, see Materials and methods for a detailed description of the optical setup). The SLM allowed projecting holographic oval shapes covering the soma of target neurons (*Figure 2C*). In urethane-anaesthetized mice, we then performed two-photon-targeted juxtasomal recordings from stCoChR-positive neurons in order to record their supra-threshold electrical activity (*Figure 2C and D*) before, during, and after holographic stimulation of the recorded neurons (shape diameter: 10–15 µm; average power *per* neuron: ≤30 mW; stimulation duration: ≤100 ms; $\lambda_{stim}$ = 920 nm). We observed a significant increase in the firing rate of the recorded neuron during the stimulation period (*Figure 2E*). Neuronal responses to holographic two-photon stimulation increased with the average power of stimulation (*Figure 2F*). Importantly, the firing frequency increase during two-photon illumination was significantly larger for stCoChR-positive neurons than for stChR2-expressing cells previously recorded (*Forli et al., 2018*) under similar experimental conditions (unpaired Student's t-test, p=3.5E-3, for 10 mW average power; Mann-Whitney test, p=4.6E-5, for 30 mW average power, $\lambda_{stim}$ = 920 nm, *Figure 2F*). To evaluate the temporal precision of stimulation, we computed the action potential (AP) probability as a function of illumination duration: at all

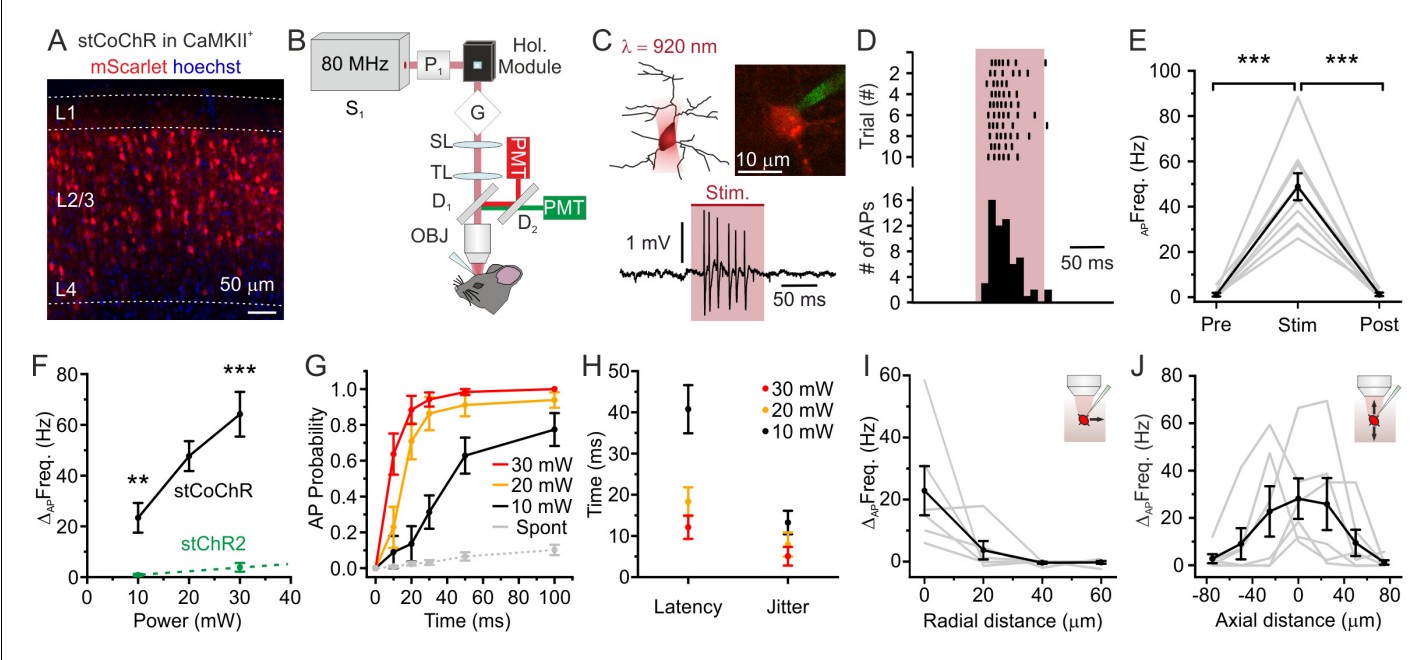

**Figure 2.** High-efficiency two-photon holographic stimulation of neurons expressing the large-conductance opsin stCoChR in vivo. (A) Confocal image showing cortical principal cells expressing stCoChR (CoChR-Kv2.1-P2A-mScarlet). (B) Schematic of the optical set up. (C) A holographic oval shape projected on the target neuron (top left), which is recorded in the juxtasomal electrophysiological configuration (top right). Representative trace from one stimulated neuron (bottom panel). The pink bar indicates the stimulation period (stimulation power: 20 mW; illumination duration: 100 ms). (D) Raster plot of holographic stimulation-induced firing activity across consecutive sweeps (top graph) in the cell shown in (C). The corresponding AP time histogram is shown in the bottom panel. (E) Frequency of APs before (Pre), during (Stim), and after (Post) the holographic two-photon stimulation. Individual experiments are shown in gr, the average across experiments in black (stimulation power: 20 mW). One-way repeated measure ANOVA with post hoc Bonferroni correction, p=1.3E-5, n = 10 cells from four mice. (F) Change in AP frequency as a function of the average stimulation power. Black indicates results for cells expressing stCoChR (n = 10 neurons from four animals). Green indicates results for cells expressing stChR2 (n = 8 neurons from four animals). For 10 mW average power, unpaired Student's *t*-test, p=3.5E-3; for 30 mW average power, Mann-Whitney test, p=4.6E-5, $\lambda_{stim}$ = 920 nm. Data in green are taken from *Forli et al., 2018*. (G) Probability of discharging one or more AP as a function of time during holographic stimulation, n = 10 cells from four animals. Red, yellow, and black indicate different average stimulation powers. Gr indicates the spontaneous AP probability in the absence of holographic stimulation. (H) Latency to first AP and jitter of first AP in holographic stimulation experiments on cells expressing stCoChR, n = 10 cells from four mice. Average stimulation power is color coded. (I) Firing frequency increase *vs* displacement of the excitation volume in the radial direction during holographic illumination of L2/3 cells expressing stCoChR. (J) Firing frequency increase *vs* displacement of the excitation volume in the axial direction during holographic illumination of L2/3 cell expressing stCoChR. For (I and J), n = 6 cells from five mice. Stimulation power: 15 mW.

The online version of this article includes the following source data for figure 2:

**Source data 1.** Source data for *Figure 2*.

tested average powers (10, 20, and 30 mW), one or more spikes could be recorded in a time window of ≤100 ms with high probability (*Figure 2G*). The latency to first spike and the jitter depended on the illumination power (*Figure 2H*). The spatial resolution of holographic stimulation in the radial and axial (dorso-ventral) direction was measured by progressively shifting the illumination volume in the corresponding directions (*Figure 2I,J*) while recording the response of the illuminated neuron with juxtasomal recording. The half-response distance was 4.4 ± 3.8 μm in the radial direction, 17.8 ± 4.4 μm in the axial up (dorsal) direction, and 23.8 ± 4.3 μm in the axial down (ventral) direction (n = 6 neurons from five mice). Taken together, these results demonstrate that stCoChR allows higher efficiency two-photon holographic stimulation compared to previously characterized blue-light-sensitive opsins in vivo (*Chen et al., 2019*; *Forli et al., 2018*), while maintaining high spatial resolution.

## Two-photon holographic stimulation of stCoChR-expressing neurons with low average power in vivo

The extended illumination approach that we used to stimulate neurons spreads the energy of the laser pulse across the cell body volume of the stimulated targets. This naturally decreases the energy density and the risk for instantaneous non-linear photodamage during illumination, allowing higher energies *per* pulse to be used. We therefore investigated the possibility to stimulate stCoChR-expressing neurons using an ultrafast pulsed laser with high-energy *per* pulse and low repetition rate. The laser was coupled with a non-collinear optical parametric amplifier (NOPA) which allowed to stimulate cells around the peak wavelength ($\lambda_{stim}$ = 920 nm) of the opsin's two-photon action spectrum (*Shemesh et al., 2017*). The NOPA output was directed to the holographic module through a switching mirror, as shown in *Figure 3A* (see Materials and methods, for a detailed description of the setup). We performed two-photon-targeted juxtasomal recordings in vivo and compared the stimulation response of the same stCoChR-positive neurons to high (80 MHz) or low (1 MHz) repetition rate (stimulus pulse duration: 100 ms; excitation shape diameter:10–15 µm; average stimulation power at 80 MHz: 10–30 mW; average stimulation power at 1 MHz: 1–5 mW; *Figure 3B*). We found a significant increase (Wilcoxon signed rank test, p=0.031, n = 6 cells from two mice) in the neuronal response using low-repetition rate laser sources compared to using high-repetition rate laser sources (*Figure 3C and D*). The AP probability as a function of illumination duration, the latency to first spike, and the jitter of the first spike for stimulation at low repetition rate at various average powers are reported in *Figure 3E–F*. Using the low repetition rate laser, we also stimulated with train of short pulses (pulse duration: 10 ms; number of pulses: 5) at different frequency (20–50 Hz, *Figure 3—figure supplement 1*). We found that the probability of eliciting at least one AP raised from 20 Hz to 40 Hz and then tended to decrease at 50 Hz. This is compatible with the depolarization induced by the stimulus *n* partially summing up to the depolarization induced by the pulse *n-1*. At lower stimulation frequencies (e.g. 20 Hz) summation leads to increased probability of spiking in the *n* stimulus. At higher stimulation frequencies (e.g. 50 Hz), summation leads to decreased spiking rate, likely due to larger depolarization during the summation process and consequent inactivation of voltage-gated conductances.

We extended the comparison between stimulation at high repetition rate and low repetition rate to other neuron types. We expressed stCoChR in a subpopulation of cortical interneurons – the somatostatin-positive (SST$^+$) cells – via AAV injections *Figure 3—figure supplement 2A*. We repeated the two-photon holographic stimulation experiments paired with imaging-guided juxtasomal recordings described above using high (80 MHz) or low (1 MHz) repetition rate laser sources (*Figure 3—figure supplement 2B*). Similarly to what we observed on L2/3 principal cells, we found that the increase in firing rate as a function of average illumination power was steeper when using low (1 MHz) repetition rate stimulation compared to using high (80 MHz) repetition rate stimulation (*Figure 3—figure supplement 2C*). As a consequence, the increase in firing rate *per* mW was significantly higher at 1 MHz stimulation compared with 80 MHz stimulation (paired Student's *t*-test, p=3E-4, n = 8 cells from four mice, *Figure 3—figure supplement 2D*). Using 3 mW and 5 mW average stimulation power, one or more spikes could be recorded in ≤30 ms with high probability (*Figure 3—figure supplement 2E*). The latency to first spike is displayed in *Figure 3—figure supplement 2F* and *Figure 3—figure supplement 3*.

Considering the slow decay time of the opsin photocurrent (in the ms range), we reasoned that the repetition rate of the stimulation laser could be further decreased below 1 MHz, while maintaining high-efficiency opsin stimulation and lowering average stimulation power. We explored this possibility by using the *pulse picking* option of our laser + NOPA system, which decreases the laser repetition rate and keeps the energy *per* pulse constant (thus decreasing the average stimulation power (*Figure 4A*)). We holographically stimulated stCoChR-expressing L2/3 principal cells (energy *per* pulse under the objective: 5 nJ; stimulus duration: 100 ms; shape diameter: 10–15 µm) while progressively decreasing repetition rate from 1 MHz to 50 kHz (*Figure 4B*). As shown in *Figure 4C*, we found that AP probability non-linearly decreased with the repetition rate. A similar AP probability during illumination was observed when stimulating at 1 MHz repetition rate with 1 mW average power and lower energy *per* pulse (1 nJ) and when stimulating at 50 kHz repetition rate with 0.25 mW average power and higher energy *per* pulse (5 nJ) (*Figure 4D*). Similar findings were observed in SST interneurons expressing stCoChR (*Figure 4—figure supplement 1*). These results

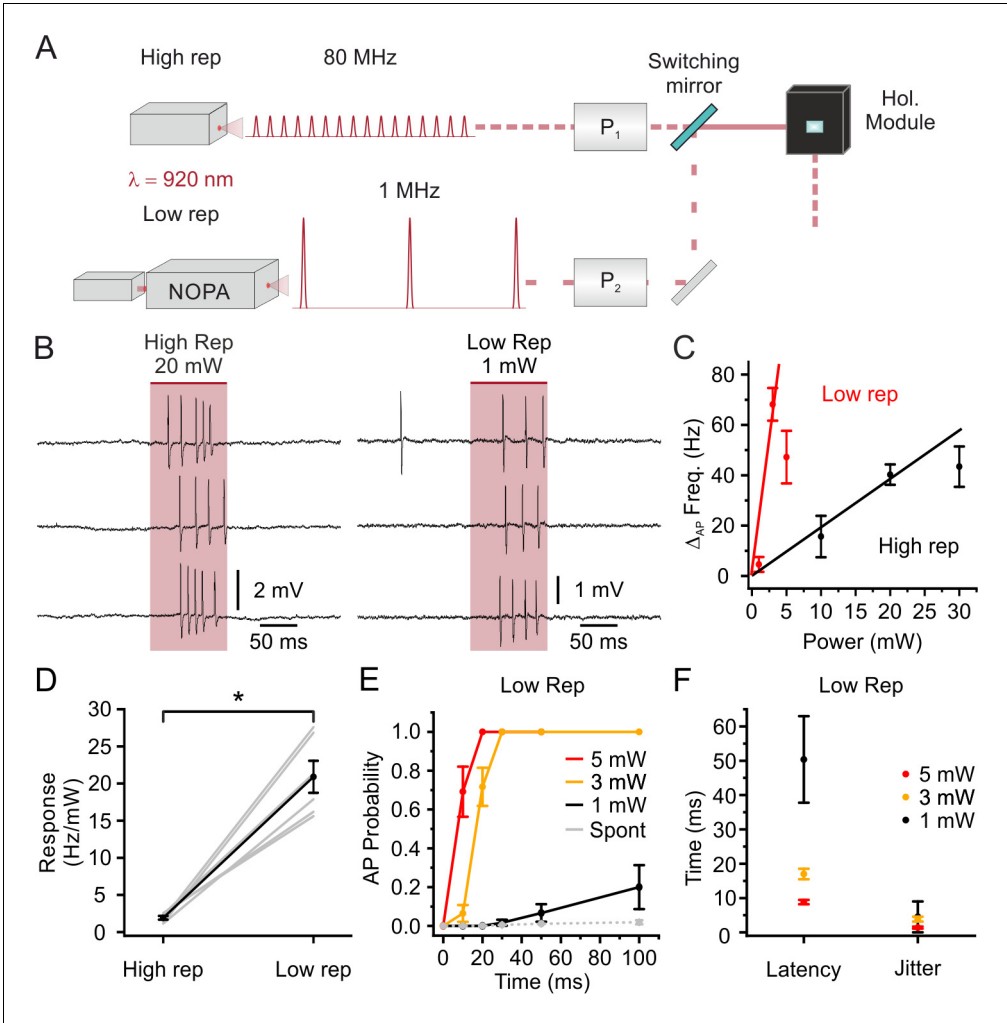

**Figure 3.** Stimulation with low repetition rate laser sources decreases the average power delivered to the sample while maintaining high stimulation efficiency: principal neurons. (**A**) Schematic of the experimental configuration: one high (80 MHz) and one low (1 MHz) repetition rate laser sources are alternated to stimulate the same electrophysiologically recorded cell. Both laser sources are tuned at 920 nm. $P_1$, Pockels cell 1; $P_2$, Pockels cell 2; NOPA, Non- collinear Optical Parametric Amplifier. (**B**) Representative traces from the same recorded L2/3 pyramidal neuron in vivo showing the effect of holographic stimulation using the high (left, average power: 20 mW) and the low (right, average power: 1 mW) repetition rate laser. (**C**) Change in AP frequency as a function of average stimulation power. Black indicates results using the high repetition rate laser and red indicates results using the low repetition rate laser. The red and black lines are fitting the values obtained with non-saturating stimulation power (1 and 3 mW for low repetition, 10 and 20 mW for high repetition, respectively). In this as well in the other panels of this figure, n = 6 cells from two mice. (**D**) Neural response in terms of AP frequency increase *per* mW of delivered average laser power in the case of stimulation with the high repetition rate laser (High rep) and low repetition rate laser (Low rep). Wilcoxon signed rank test, p=0.031. (**E**) AP probability as a function of duration in holographic stimulation experiments using low repetition rate on L2/3 pyramidal neurons expressing stCoChR. Red, yellow, and black indicate different average stimulation powers. Grey indicates the spontaneous AP probability in the absence of holographic stimulation. (**F**) Latency to first AP and jitter of first AP in holographic stimulation experiments using low repetition rate on L2/3 pyramidal neurons expressing stCoChR. Average stimulation power used in the experiments is indicated with the color code.

The online version of this article includes the following source data and figure supplement(s) for figure 3:

**Source data 1.** Source data for *Figure 3*.
**Figure supplement 1.** Spike control in stCoChR-expressing neurons at different stimulation frequencies.
**Figure supplement 1—source data 1.** Source data_for *Figure 3—figure supplement 1*.
**Figure supplement 2.** Stimulation with low repetition rate laser sources decreases the average power delivered to the sample while maintaining high stimulation efficiency: somatostatin (SST)-positive interneurons.
*Figure 3 continued on next page*

*Figure 3 continued*

**Figure supplement 2—source data 1.** Source data_for *Figure 3—figure supplement 2*.
**Figure supplement 3.** Latency and jitter decrease with the average power of low repetition rate stimulation.
**Figure supplement 3—source data 1.** Source data_for *Figure 3—figure supplement 1*.

demonstrate highly efficient two-photon holographic stimulation of stCoChR with low average power *per* cell (0.25–5 mW, depending on the laser repetition rate) and energy densities well below photodamage thresholds (*Charan et al., 2018*; *Chen et al., 2019*).

## Crosstalk between imaging and stimulation with stCoChR in vivo

Given the high efficiency of two-photon holographic stimulation shown by stCoChR, we asked whether raster imaging at a long wavelength (~1100 nm), typically used to image red-shifted functional indicators, would lead to significant increase in the firing rate of stCoChR-expressing neurons in vivo (crosstalk). We thus measured the supra-threshold responses of stCoChR-expressing L2/3 pyramidal neurons using juxtasomal electrophysiological recordings while performing two-photon raster scanning in vivo (*Figure 5*). Raster scanning was performed at 11 Hz and 30–35 mW average power in a rather small (dimension: $161.4 \times 161.4$ μm$^2$) region of interest (ROI, *Figure 5A*). We first tuned the imaging laser at $\lambda_{scanning}$ = 1100 nm, the wavelength that would have been used if a red-shifted functional indicator were expressed (*Figure 5B*, top panel). We found that raster scanning under these conditions did not significantly affect the firing activity of L2/3 neurons expressing stCoChR (*Figure 5C*, left). As an important control, we tuned the imaging laser to 920 nm (the wavelength used for holographic stimulation) and we found that raster scanning the same FOV at 920 nm and moderate power (laser power: 30 mW) significantly increased the spiking activity of stCoChR-expressing neurons (*Figure 5B and C*, right). Co-expression of another protein (e.g. jRCaMP1a) could affect the expression levels of stCoChR. We thus repeated the crosstalk experiment in vivo in individual neurons co-expressing stCoChR and jRCaMP1a while simultaneously performing functional imaging and juxtasomal electrophysiological recording (*Figure 5D–F*). Similarly to what we observed in cells expressing only stCoChR, we found that raster scanning at 1100 nm (imaging power: 30–35

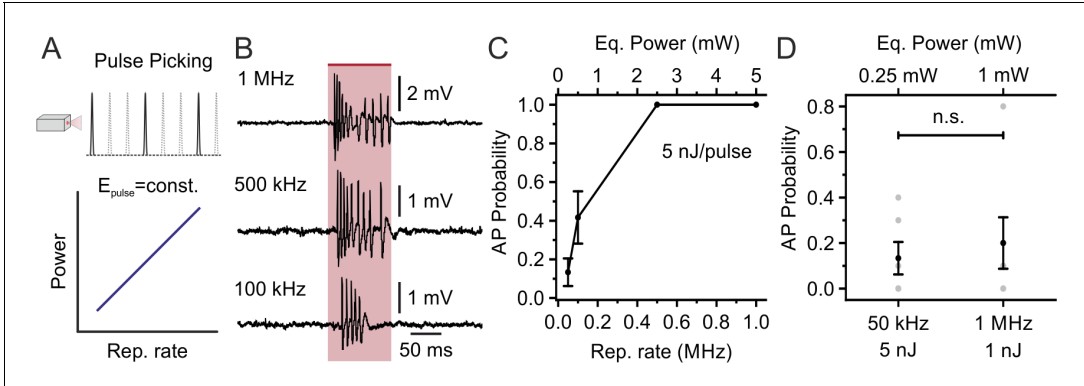

**Figure 4.** Decreasing the laser repetition rate to <1 MHz allows holographic stimulation of stCoChR-expressing cells with <1 mW average power *per* cell: principal neurons. (**A**) The laser repetition rate is decreased using pulse picking (top panel). With this approach, the average power delivered to the sample linearly increases with the repetition rate (bottom panel). (**B**) Representative traces from the same electrophysiologically recorded L2/3 pyramidal neuron stimulated at different repetition rates (from top to bottom panel: 1 MHz, 500 kHz, 100 kHz; energy *per* pulse: 5 nJ). (**C**) Probability of discharging one or more AP as a function of the laser repetition rate for principal neurons expressing stCoChR in vivo (illumination duration: 100 ms). n = 6 cells from two mice. (**D**) AP probability obtained at 50 kHz and 1 MHz repetition rates, but delivering lower average power (0.25 mW *vs* 1 mW) and higher energy *per* pulse (5 nJ *vs* 1 nJ) at 50 kHz compared to 1 MHz. n = 6 cells from two mice; Wilcoxon signed rank test p=0.88.

The online version of this article includes the following source data and figure supplement(s) for figure 4:

**Source data 1.** Source data for *Figure 4*.
**Figure supplement 1.** Decreasing the laser repetition rate to <1 MHz allows holographic stimulation of stCoChR-expressing cells with <1 mW average power *per* cell: SST-positive interneurons.
**Figure supplement 1—source data 1.** Source data for *Figure 4—figure supplement 1*.

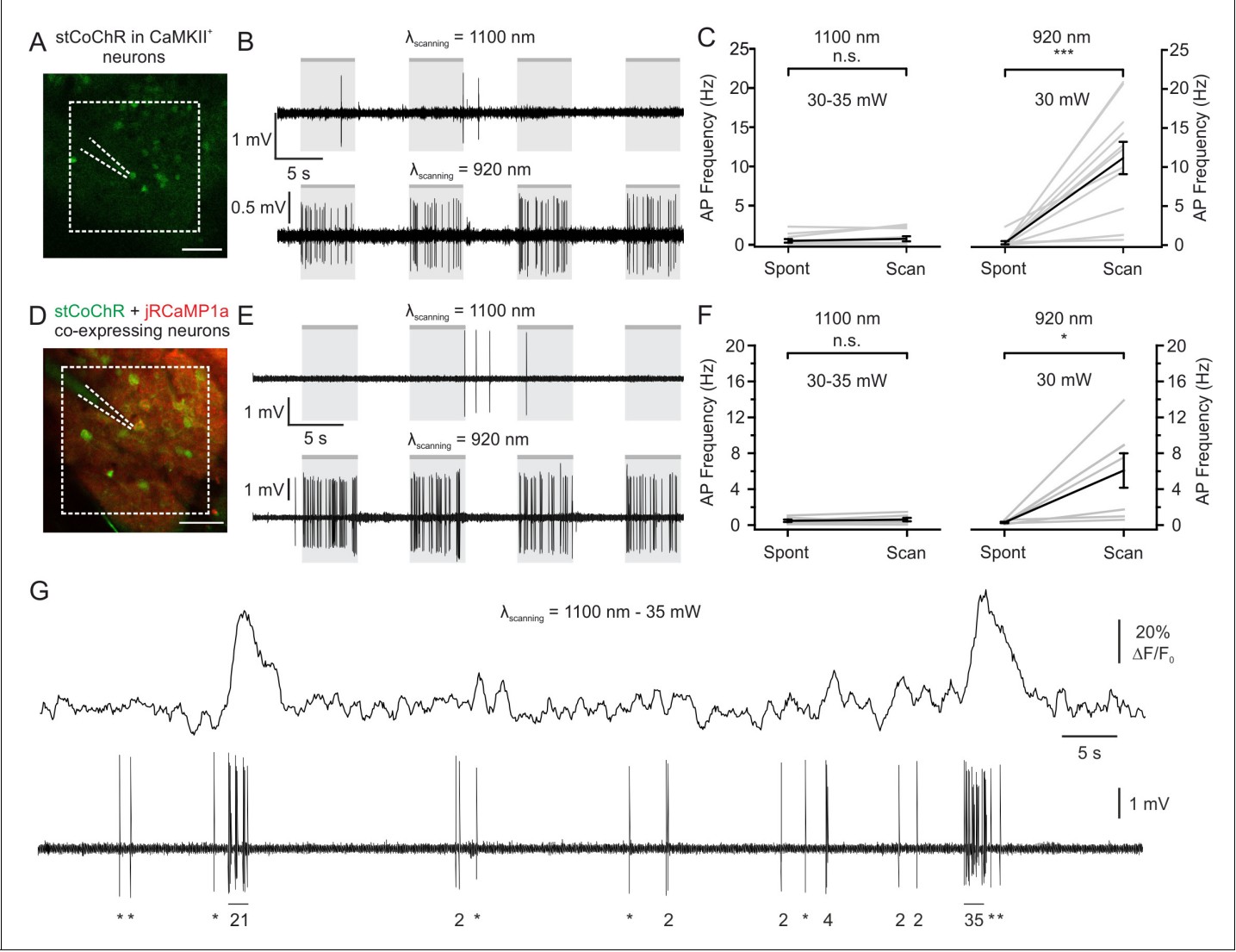

**Figure 5.** Limited crosstalk between imaging and photostimulation using blue-light-sensitive opsins and red-shifted indicators. (**A**) Two-photon image of L2/3 principal cells expressing stCoChR in vivo (CoChR-Kv2.1-P2A-NLS-eGFP). One opsin-positive neuron was recorded with a glass pipette (dashed lines) in the juxtasomal configuration while two-photon raster scanning inside the indicated area (dashed white box, $161.4 \times 161.4 \ \mu m^2$) was performed (imaging frame rate: 11 Hz). (**B**) Traces recorded in the juxtasomal electrophysiological configuration from one stCoChR-expressing neuron during epochs (gr boxes) of two-photon raster scanning at wavelength 1100 nm (top) and 920 nm (bottom). Laser power: 30 mW. (**C**) Average AP frequency during epochs of spontaneous activity (Spont) and during raster scanning (Scan). The left panel shows results when scanning was performed at $\lambda = 1100$ nm. The right panel displays the results when scanning was done at $\lambda = 920$ nm. n = 11 cells from six mice; Wilcoxon signed rank test, p=0.31 for $\lambda = 1100$ nm; Wilcoxon signed rank test, p=9.8E-4 for $\lambda = 920$ nm. (**D**) Same as (**A**), but for cells co-expressing stCoChR (CoChR-Kv2.1-P2A-NLS-eGFP) and jRCaMP1a. (**E**) Same as (**B**), but from a cell co-expressing stCoChR and jRCaMP1a. (**F**) Same as (**C**), but for neurons co-expressing stCoChR and jRCaMP1a. n = 7 cells from three mice; Wilcoxon signed rank test, p=0.21 for $\lambda = 1100$ nm; Wilcoxon signed rank test, p=0.02 for $\lambda = 920$ nm. (**G**) Representative trace from a L2/3 pyramidal neuron co-expressing stCoChR and jRCaMP1a. AP firing was recorded in the juxtasomal configuration (bottom) while two-photon raster scanning was performed (top). Imaging frame rate and FOV dimensions as in A; average imaging power, 35 mW. The number of recorded APs is reported below the electrophysiological trace (single APs are indicated with an asterisk).

The online version of this article includes the following source data and figure supplement(s) for figure 5:

**Source data 1.** Source data for *Figure 5*.

**Figure supplement 1.** Crosstalk between imaging and photostimulation: SST-positive interneurons Related to *Figure 5*.

**Figure supplement 1—source data 1.** Source data for *Figure 5—figure supplement 1*.

**Figure supplement 2.** $\Delta F/F_0$ of spontaneous jRCaMP1a events did not significantly increase with imaging average power.

**Figure supplement 2—source data 1.** Source data for *Figure 5—figure supplement 2*.

**Figure supplement 3.** Crosstalk between imaging and photostimulation at 50 mW imaging average power.

**Figure supplement 3—source data 1.** Source data for *Figure 5—figure supplement 3*.

mW) did not alter the spontaneous firing rate of imaged neurons (*Figure 5E–F*). Moreover, raster scanning at 920 nm caused a large and significant increase in the neuron's spontaneous firing rate (*Figure 5E–F*). Importantly, we verified that the average imaging power used to estimate the cross-talk (between 30 mW and 35 mW) was sufficient to detect jRCaMP1a fluorescence transients associated with spontaneous firing of APs (*Figure 5G*), similarly to previous reports (*Dana et al., 2016*; *Forli et al., 2018*). We also performed a similar experiment with SST-positive interneurons expressing stCoChR (*Figure 5—figure supplement 1*). In agreement with our findings on principal cells, scanning stCoChR-positive interneurons at 1100 nm did not significantly raise the firing rate of recorded cells, while scanning at 920 nm did (*Figure 5—figure supplement 1B* and *Figure 5—figure supplement 1C*). Finally, in principal cells co-expressing jRCaMP1a and stCoChR further increasing the average imaging power to 50 mW did not increase the $\Delta F/F_0$ of calcium events in cells expressing jRCaMP1a (*Figure 5—figure supplement 2*), while it led to small but significant increase in the AP frequency induced by raster scanning in both cells expressing stCoChR and cells expressing stCoChR +jRCaMP1 a (*Figure 5—figure supplement 3*). Overall, these results demonstrate that imaging at longer wavelengths (~1100 nm) with average power allowing detection of calcium events (30–35 mW) does not significantly perturb the supra-threshold activity of stCoChR-positive principal neurons and interneurons.

## Simultaneous two-photon imaging of jRCaMP1a and two-photon holographic stimulation of stCoChR

We finally combined two-photon holographic stimulation of stCoChR with two-photon imaging of the red-shifted calcium indicator jRCaMP1a (*Dana et al., 2016*). To this aim, we used one 80 MHz laser source tuned at $\lambda_{imaging}$ = 1100 nm for imaging and one low repetition rate laser ($\leq$1 MHz) tuned at $\lambda_{stim}$ = 920 nm for stimulation (*Figure 6A*, see Materials and methods for details). We co-expressed jRCaMP1a and stCoChR coupled to a nucleus-localized eGFP (CoChR-Kv2.1-P2A-NLS-eGFP) in L2/3 pyramidal neurons of the mouse cortex (*Figure 6B*), and we programmed the holographic module to project extended shapes covering the cell bodies of target neurons. We simultaneously stimulated ensembles of multiple neurons while imaging their activity (stimulus duration: 200 ms; average stimulus energy *per* pulse under the objective: 7 nJ; laser repetition rate: 0.1–1 MHz; number of stimulated neurons: 5–30). We observed that consecutive holographic stimulation of stCoChR-expressing neurons typically evoked reliable fluorescence transients in the targeted cells (*Figure 6C*). We investigated the relationship between the amplitude of evoked fluorescence transients and the repetition rate of the stimulation laser, at constant energy *per* pulse (*Figure 6D–E*). We found that fluorescence transients in the stimulated cell could be evoked with low power (0.7 mW *per* cell at 0.1 MHz repetition rate, *Figure 6D*) and that increasing the repetition rate non-linearly increased the amplitude of the evoked fluorescence transients (*Figure 6E*). We performed holographic two-photon stimulation and imaging in experiments in individual cells in which we monitored the firing activity with juxtasomal electrophysiological recordings (*Figure 6F–G*). We observed that jRCaMP1a transients induced by the used photostimulation protocol were associated with the firing of 6.7 ± 1.9 APs (range 3–14, n = 5 cells from three mice), in agreement with what we observed in cells expressing only stCoChR (*Figure 2C–E*). Importantly, in cells expressing only jRCaMP1a the same stimulation protocol did not increase AP frequency (*Figure 6—figure supplement 1*). We also compared spontaneous and photostimulation-evoked jRCaMP1a transients (*Figure 6—figure supplement 2*). We found that the $\Delta F/F_0$ of spontaneous events corresponding to the discharge of few APs were smaller than the $\Delta F/F_0$ of photostimulation-evoked jRCaMP1a transients, in agreement with the observation that 200 ms of photostimulation evoked trains of 3–14 APs in stimulated neurons (*Figure 6—figure supplement 2*). The decay kinetic of spontaneous and photostimulation-evoked jRCaMP1a transients were not significantly different (*Figure 6—figure supplement 2*).

Altogether, these results demonstrate stCoChR is a highly efficient blue-shifted opsin that can be used in combination with red-shifted calcium indicators to perform all-optical circuit interrogations with low average power delivery *per* cell and minimal crosstalk between imaging and photostimulation.

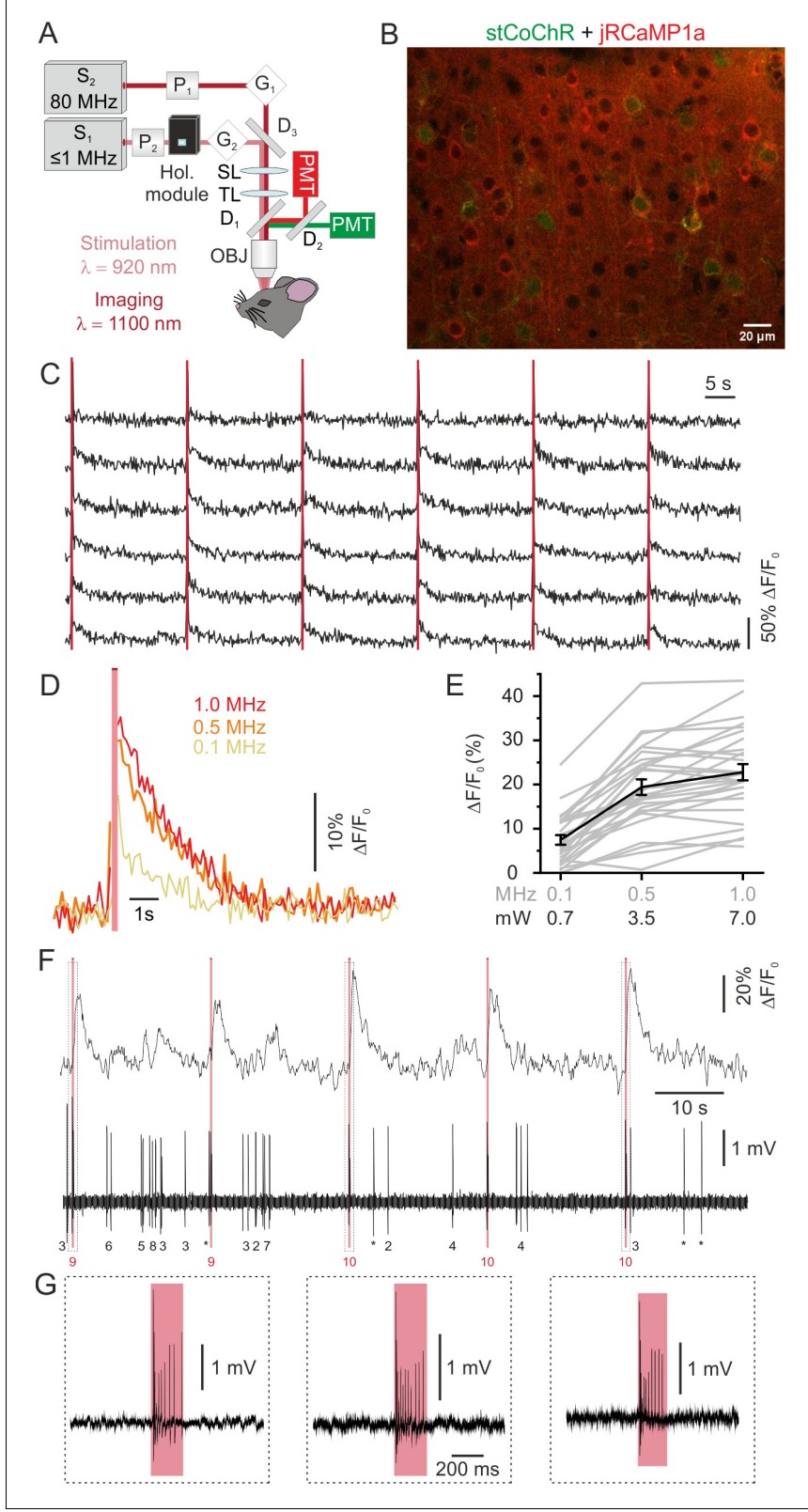

**Figure 6.** All-optical imaging and high-efficiency manipulation combining jRCaMP1a and stCoChR. (**A**) Schematic of the optical system. Two laser sources tuned at different wavelengths are combined in the same set up for simultaneous imaging and optogenetic manipulation. (**B**) Confocal image showing cortical L2/3 neurons expressing jRCaMP1a (red) and stCoChR (CoChR-Kv2.1-P2A-NLS-eGFP, green). (**C**) Representative traces showing

*Figure 6 continued on next page*

*Figure 6 continued*

the response of an ensemble of jRCaMP1a- and stCoChR-expressing neurons to successive holographic stimulations (pink bars) in vivo. Periods of stimulation are blanked. Traces from six stimulated neurons are shown in the panel. Average imaging power range, 30–35 mW; frame rate, 11 Hz. (D) Representative traces showing average jRCaMP1a signals in response to holographic stimulation at different laser repetition rates (from 100 kHz to 1 MHz; average stimulation energy *per* cell: 7 nJ *per* pulse, 200 ms illumination; n = 28 cells from two mice, six consecutive stimulations *per* cell). (E) $\Delta F/F_0$ of jRCaMP1a signals as a function of the repetition rate (or the equivalent average power *per* cell) of the stimulating laser, n = 28 cells from two mice. Mean values (of six consecutive stimulations) from individual cells are shown in grey, the average across experiments in black. (F) Representative trace showing jRCaMP1a fluorescence (top) and the corresponding juxtasomal electrophysiological recording from one cell co-expressing jRCaMP1a and stCoChR. The pink bar indicates the stimulation period (average stimulation power, 5 mW; repetition rate, 1 MHz; stimulus duration, 200 ms). The number of recorded APs is reported below the electrophysiological trace (single APs are indicated with an asterisk; the number of AP during holographic stimulation is shown in red). (G) The juxtasomal electrophysiological trace during holographic stimulation is shown at an expanded time scale and shows stimulation-induced AP trains.

The online version of this article includes the following source data and figure supplement(s) for figure 6:

**Source data 1.** Source data for *Figure 6*.
**Figure supplement 1.** Photostimulation of neurons expressing only jRCaMP1a did not increase the neuron's firing rate.
**Figure supplement 1—source data 1.** Source data for *Figure 6—figure supplement 1*.
**Figure supplement 2.** Spontaneous and stimulation-evoked jRCaMP1a transients.
**Figure supplement 2—source data 1.** Source data for *Figure 6—figure supplement 2*.

## Discussion

All-optical interrogation of neural circuits is increasingly recognized as a fundamental tool to causally investigate the codes that neuronal circuits use to drive behavior. By combining two-photon excitation of functional indicators and optogenetic actuators, it has now been possible to simultaneously image and manipulate neural networks with near single-cell resolution in the intact brain of behaving animals (*Carrillo-Reid et al., 2019*; *Gill et al., 2020*; *Jennings et al., 2019*; *Marshel et al., 2019*). Ideally, optogenetic actuators should require minimal light power for neuronal excitation, allowing efficient stimulation of large numbers of neurons while minimizing heating (*Picot et al., 2018*). Moreover, they should display no activation by the light wavelength used for imaging, that is limited crosstalk between imaging and photostimulation. Here, we demonstrate that the large-conductance blue-light-sensitive soma-targeted opsin, stCoChR, in combination with high-energy ultrafast laser sources tuned at the opsin's peak absorption wavelength, requires more than one order of magnitude lower average power *per* cell compared to previously characterized blue-light-sensitive soma-targeted opsins in vivo. Moreover, imaging conditions commonly used to monitor red-shifted functional indicators do not result in significant suprathreshold activation of stCoChR-expressing neurons.

In a previous study, Shemesh et al. demonstrated temporally precise stimulation of neurons in vitro using a different somatic variant of CoChR (soCoChR, *Shemesh et al., 2017*). In that study, somatic targeting of CoChR was achieved through a sequence from the kainate receptor (KA2). The resulting soCoChR yielded decreased somatic photocurrents with respect to the unmodified CoChR. In this manuscript, we used a different targeting motif, taken from the Kv2.1 channel (*Baker et al., 2016*; *Chettih and Harvey, 2019*; *Forli et al., 2018*; *Mahn et al., 2018*; *Mardinly et al., 2018*; *Marshel et al., 2019*), to enrich CoChR expression in the somatic compartment. While Shemesh et al. did screen for soma-targeting with the Kv2.1 motif in their study, they conducted this initial screen using dissociated cultured neurons. Our previous work has demonstrated that the efficiency of soma-targeting with the Kv2.1 motif is greatly improved in vivo compared with the dissociated culture (*Mahn et al., 2018*), which might explain its apparent inferior performance in the Shemesh et al. study. Furthermore, Shemesh et al. added the Kv2.1 targeting sequence following the CoChR-GFP sequence, whereas our construct contains the Kv2.1 targeting sequence immediately following the CoChR coding sequence, positioning it closer to the membrane as it is naturally located in the Kv2.1 channel sequence. We found that while the photocurrent elicited by stCoChR under full-field illumination is similar to the photocurrent elicited by unmodified CoChR, a higher fraction of that

current originated from the soma in stCoChR-expressing cells (*Figure 1*). The reduction in extra-somatic photocurrents was consistent in stCoChR-expressing cells, while it was only observed in soCoChR-expressing cells under very high light power. Our results suggest that the overall level of expression and efficiency of targeting to the plasma membrane are similar for CoChR and stCoChR, while stCoChR molecules are better retained at the proximal membrane. Thus, stCoChR achieves soma restriction without compromising photocurrent. Although the molecular mechanisms leading to these results remain to be investigated, we found that stCoChR led to very efficient two-photon neuronal excitation which required an average power of 0.25–5 mW *per* cell.

Limiting the crosstalk between imaging and photostimulation is a critical aspect of all-optical methods. Red-shifted opsins are generally prone to unwanted activation during imaging of blue-light-sensitive indicators (*Forli et al., 2018*; *Venkatachalam and Cohen, 2014*), because of the intrinsic absorption properties of the retinal group at shorter wavelengths. Previous work demon-strated that combining blue-light-sensitive opsins (e.g. ChR2, GtACR2) with red-shifted indicators (jRCaMP1a) is a valid choice to effectively reduce this form of crosstalk (*Forli et al., 2018*). Impor-tantly, despite the fact that stCoChR has a much larger photocurrent compared to ChR2 (*Klapoetke et al., 2014*), we show here that suprathreshold activation of stCoChR-expressing neu-rons during imaging at red-shifted wavelengths is not significant at imaging power <35 mW (*Fig-ure 5* and *Figure 5—figure supplement 1*). This allowed us to combine high-energy two-photon holographic illumination of stCoChR at around the peak of its absorption spectrum (~920 nm, [*Shemesh et al., 2017*]) with concurrent two-photon imaging of jRCaMP1a at 1100 nm (around the peak of absorption of this red-shifted indicator), achieving minimal cross activation. So far, efficient two-photon excitation of blue-shifted opsins has largely been constrained by the limited availability of high-energy pulsed systems tuned at ~920 nm. The recent accessibility of ready-to-use optical-parametric amplifiers makes it now possible to generate ultrafast high-energy pulses at a tunable wavelength in that spectral range and at a variable repetition rate (0.1–1 MHz). We used one such system to stimulate, for the first time, stCoChR at high energy around its two-photon absorption peak (920 nm, [*Shemesh et al., 2017*]), thus minimizing energy waste and activation of red-shifted fluorescent indicator. Furthermore, stimulation at 920 nm reduces tissue heating (*Podgorski and Ranganathan, 2016*) that is, in contrast, higher at the longer wavelengths (>1000 nm) commonly used to excite red-shifted opsins. Previous work showed that under the illumination conditions that we are using in our study (energy per pulse, ≤7 nJ), thermal effects dominate over non-linear photo-damage effects (*Charan et al., 2018*; *Picot et al., 2018*) and that lowering the laser repetition rate is advantageous for two-photon excitation of the retinal group (*Palczewska et al., 2020*). Moreover, previous studies showed that energies up to 60–200 nJ per pulse can be used to activate opsins without causing retinal bleaching and cell damage (*Chen et al., 2019*; *Gill et al., 2020*; *Mardinly et al., 2018*). Thus, it is conceivable that the peak energy could be further increased com-pared to the one used in our study, while still preserving opsin functionality.

The soma-restriction method described in this study effectively increased the spatial resolution of photo-stimulation (*Figure 1D,E*). However, the axial spread of the illumination profile combined with the high sensitivity of stCoChR may cause significant activation of non-target cells located above and below the focal plane (*Figure 2J*). This is not a major issue for single layered neural targets (e.g. mitral cell layer in the olfactory bulb, [*Gill et al., 2020*]) or in the case of sparse labeling, but it may compromise the interpretation of photostimulation experiments in denser cell layers, (e.g. neocorti-cal layers). The axial extension of the illumination profile can be reduced using temporal focusing (*Chen et al., 2019*). Spatial resolution could also be further increased by developing more effective soma-restriction motifs.

To reduce crosstalk between imaging and photostimulation, we imaged red-shifted indicators while stimulating blue-light-sensitive opsins. However, red-shifted indicators such as jRCaMP1a *Dana et al., 2016*; *Forli et al., 2018* have lower accuracy in detecting single or few APs compared to green indicators (*Chen et al., 2013*). Our method will benefit from future improvement leading to more efficient red-shifted indicators. Besides decreasing crosstalk, the choice of red-shifted indica-tors may facilitate imaging in deeper regions by using longer wavelengths, which are less sensitive to scattering (*Helmchen and Denk, 2005*). Moreover, the expression of red-shifted indicators is sta-ble over long time windows (*Dana et al., 2016*) and this may facilitate chronic imaging experiments. Finally, holographic stimulation at 920 nm may decrease tissue heating which is higher at the longer wavelengths (λ = 1040 nm) *Podgorski and Ranganathan, 2016* used to stimulate red-shifted opsins

(e.g. C1V1) (*Carrillo-Reid et al., 2017*; *Packer et al., 2012*; *Packer et al., 2015*; *Prakash et al., 2012*; *Rickgauer et al., 2014*).

Compared to scanning methods with diffraction-limited-spots (*Carrillo-Reid et al., 2016*; *Carrillo-Reid et al., 2019*; *Chettih and Harvey, 2019*; *Marshel et al., 2019*; *Packer et al., 2015*; *Yang et al., 2018*), our extended illumination approach distributes the laser pulse energy across a much larger volume, that is the cell body volume of the stimulated neuron. Thus, the instantaneous energy density and the consequent risk of non-linear photodamage are decreased, allowing higher energy *per* pulse to be used. We exploited this advantage and compared light-evoked responses of the same stCoChR-expressing neurons to stimulation with low energy *per* pulse and high repetition rate (80 MHz) or high energy *per* pulse and low repetition rate (1 MHz). Previous studies *Carrillo-Reid et al., 2019*; *Chaigneau et al., 2016*; *Gill et al., 2020*; *Mardinly et al., 2018*; *Marshel et al., 2019*; *Ronzitti et al., 2017*; *Shemesh et al., 2017*; *Yang et al., 2018* used the direct output of a low repetition rate laser ($\lambda \sim 1040$ nm) for opsin excitation. This effectively stimulates red-shifted opsins, but not blue-light-sensitive opsins. Here, we combined a low repetition rate laser with a non-collinear optical parametric amplifier, to efficiently stimulate blue-light-sensitive opsins with high energy pulses at the wavelength corresponding to the peak of the two-photon action spectrum of CoChR ($\lambda = 920$ nm). Excitation at a low repetition rate allowed us to significantly increase the response of illuminated neurons *per* mW of delivered average power (*Figure 3* and *Figure 3—figure supplement 2*). Moreover, we reasoned that the slow kinetics of the opsins and the threshold for non-linear photodamage would allow us to further reduce the repetition rate of the excitation light source and the average power delivered to the sample. We found that the probability of eliciting suprathreshold responses during two-photon holographic excitation sublinearly decreased with the repetition rate (energy *per* pulse was 5 nJ under the objective and constant for the different repetition rates, *Figure 4* and *Figure 4—figure supplement 1*). Importantly, we found that, at 50 kHz repetition rate, stimulation of target cells could be achieved with <1 mW, more than one order of magnitude lower average power per cell compared to previously published blue-light-sensitive soma-targeted (st) opsins in vivo (*Forli et al., 2018*). Importantly, we validated these findings across both principal cells and interneurons (SST[+]), showing that two-photon holographic stimulation can be applied to cell types with different biophysical properties with similar efficacy. Altogether, the use of stCoChR in vivo, compared with other opsins including ST-ChroME, Chronos and the non-soma-targeted version of CoChR (*Chen et al., 2019*; *Mardinly et al., 2018*), allows photoexcitation with lower average power of excitation (1–5 mW at 1 MHz repetition rate, and potentially less at repetition rates < 1 MHz, *Figure 3*, *Figure 3—figure supplement 2*). These results pave the way for the simultaneous modulation of the electrical activity of large number of neurons for prolonged periods with minimal risk of tissue heating (*Picot et al., 2018*; *Podgorski and Ranganathan, 2016*).

In conclusion, we developed a new soma-targeted variant of the large-conductance opsin CoChR. Compared to previously characterized blue-light-sensitive soma-targeted opsins in vivo, stCoChR allowed comparable neuronal stimulation with more than one order of magnitude decreased average power delivered to the brain tissue and with minimal crosstalk between imaging and photostimulation. The combination of stCoChR with tuned amplified laser stimulation and red-shifted functional indicators represents a powerful alternative approach to performing high-efficiency all-optical causal investigation of neural circuits in the intact brain with minimal crosstalk between the imaging and photostimulation channels.

## Materials and methods

### Key resources table

| Reagent type (species) or resource | Designation | Source or reference | Identifiers | Additional information |
|---|---|---|---|---|
| Strain, strain background (*M. musculus*) | C57BL/6J | Charles River | IMSR Cat# JAX:000664, RRID:IMSR_JAX:000664 | |

*Continued on next page*

*Continued*

| Reagent type (species) or resource | Designation | Source or reference | Identifiers | Additional information |
|---|---|---|---|---|
| Strain, strain background (*M. musculus*) | STOCK Sst<sup>tm2.1(cre)</sup>*Zjh/J* | The Jackson Laboratory | IMSR Cat# JAX:013044, RRID:IMSR_JAX:013044 | |
| Recombinant DNA reagent | pAAV-Syn-CoChR-GFP | Addgene | Addgene plasmid #59070; http://n2t.net/addgene: 59070; RRID:Addgene_59070 | |
| Recombinant DNA reagent | pAAV-EF1α-DIO-CoChR-T2A-FusionRed-WPRE | This paper | | See Materials and methods, Section 'Cloning of stCoChR and preparation of recombinant AAV vectors' |
| Recombinant DNA reagent | pAAV-EF1α-DIO-CoChR-Kv2.1-P2A-mScarlet-WPRE | This paper | | See Materials and methods, Section 'Cloning of stCoChR and preparation of recombinant AAV vectors' |
| Recombinant DNA reagent | pAAV-EF1α-DIO-CoChR-Kv2.1-P2A-NLS-eGFP-WPRE | This paper | | See Materials and methods, Section 'Cloning of stCoChR and preparation of recombinant AAV vectors' |
| Recombinant DNA reagent | pAAV-hSynapsin-FLEX-soCoChR-GFP | Addgene | Addgene viral prep # 107712-AAV9; http://n2t.net/addgene:107712; RRID:Addgene_107712; | |
| Recombinant DNA reagent | AAV1-CaMKII0.4-Cre-SV40 | Addgene | Addgene viral prep # 105558-AAV1; http://n2t.net/addgene:105558; RRID:Addgene_105558 | |
| Recombinant DNA reagent | AAV1-hsyn-flex-NES-jRCaMP1a | Addgene | Addgene viral prep # 100853-AAV1; http://n2t.net/addgene:100853; RRID:Addgene_100853 | |
| Recombinant DNA reagent | AAV1-CAG-flex-NES-jRCaMP1a | Addgene | Addgene viral prep # 100846; http://n2t.net/addgene:100846; RRID:Addgene_100846 | |
| Recombinant DNA reagent | pGP-CMV-NES-jRCaMP1a | Addgene | Addgene plasmid # 61562; http://n2t.net/addgene: 61562; RRID:Addgene_61562 | |
| Recombinant DNA reagent | AAV1-CamKII-jRCaMP1a | This paper | | See Materials and methods, Section 'Cloning of stCoChR and preparation of recombinant AAV vectors' |
| Software, algorithm | MATLAB | Mathworks | RRID:SCR_001622; https://it.mathworks.com/products/matlab.html | |
| Software, algorithm | pClamp 10 | Molecular Devices | RRID:SCR_011323; https://www.moleculardevices.com/products/axon-patch-clamp-system/acquisition-and-analysis-software/pclamp-software-suite | |
| Software, algorithm | OriginPro2018 | OriginLab | RRID:SCR_014212; https://www.originlab.com/ | |
| Software, algorithm | GraphPad PRISM | GraphPad PRISM | RRID:SCR_002798; https://www.graphpad.com/ | |
| Software, algorithm | ImageJ/Fiji | Fiji | RRID:SCR_002285; http://fiji.sc/ | |

*Continued on next page*

*Continued*

| Reagent type (species) or resource | Designation | Source or reference | Identifiers | Additional information |
|---|---|---|---|---|
| Antibody | polyclonal rabbit anti-2A peptide primary antibody (Rabbit polyclonal) | EMD Millipore | Millipore Cat# ABS31, RRID:AB_11214282catalog # ABS31 | Diluted (1:500); production discontinued |
| Antibody | polyclonal Cy5-conjugated donkey anti-rabbit secondary antibody (Donkey polyclonal) | Jackson Immuno Research | Jackson Immuno Research Labs Cat# 711-175-152, RRID:AB_2340607catalog # 711-175-152 | Diluted (1:500) |

## Animal strains

All experiments involving animals were approved by the IIT Animal Welfare Body, by the National Council on Animal Care of the Italian Ministry of Health (authorizations #34/2015-PR, #1084/2020-PR), and by the Institutional Animal Care and Use Committee at the Weizmann Institute of Science, and carried out in accordance with the guidelines established by the European Communities Council Directive. The following mouse lines were used for this study: C57BL/6J mice (IMSR Cat# JAX:000664, RRID:IMSR_JAX:000664; Charles River, Calco, Italy and Envigo, Rehovot, Israel) and STOCK *Sst^tm2.1(cre)^Zjh*/J (IMSR Cat# JAX:013044, RRID:IMSR_JAX:013044 - called SST-cre line - Jackson Laboratory, Bar Harbor, USA). Mice were housed in individually ventilated cages under a 12 hr light:dark cycle ($\leq$5 animals *per* cage). Access to food and water was ad libitum. All the in vivo experiments were performed on urethane-anesthetized young-adult animals (3–16 weeks old, either sex), as described previously (*Forli et al., 2018*). The number of animals used for each experimental dataset is specified in the text or in the corresponding Figure legend.

## Cloning of stCoChR and preparation of recombinant AAV vectors

A CoChR expression plasmid (pAAV-Syn-CoChR-GFP) was acquired from Addgene (plasmid #59070; http://n2t.net/addgene:59070; RRID:Addgene_59070)(*Klapoetke et al., 2014*). CoChR was subcloned into a Cre-dependent AAV vector backbone under the EF1α promoter (pAAV-EF1α-DIO-WPRE, [*Sohal et al., 2009*]). Cre-dependent non-targeted CoChR (pAAV-EF1α-DIO-CoChR-T2A-FusionRed-WPRE) and Cre-dependent stCoChR (pAAV-EF1α-DIO-CoChR-Kv2.1-P2A-mScarlet/eGFP-WPRE) plasmids were then engineered using standard restriction cloning. A Cre-dependent soCoChR expression plasmid (pAAV-hSynapsin-FLEX-soCoChR-GFP) was acquired from Addgene (plasmid #107712, http://n2t.net/addgene:107712; RRID:Addgene_107712). Recombinant AAV vectors containing either CoChR variant were produced as described in *Mahn et al., 2018*. An AAV plasmid encoding the jRCaMP1a gene under the CaMKIIα promoter was generated using standard restriction cloning from the plasmid pGP-CMV-NES-jRCaMP1a (Addgene plasmid # 61562; http://n2t.net/addgene:61562; RRID:Addgene_61562). pAAV-CAMKII-jRCaMP1a was packaged as AAV serotype 1–2 viral particles (*Thalhammer et al., 2017*). AAV1-CAG-flex-NES-jRCaMP1a (Addgene viral prep # 100846; http://n2t.net/addgene:100846; RRID:Addgene_100846), AAV1-CamKII0.4-Cre-SV40 (Addgene viral prep # 105558-AAV1; http://n2t.net/addgene:105558; RRID:Addgene_105558), and AAV1-hsyn-flex-NES-jRCaMP1a (Addgene viral prep # 100853-AAV1; http://n2t.net/addgene:100853; RRID:Addgene_100853) were purchased from Addgene.

## Viral injections for in vitro experiments

To achieve sparse expression of either CoChR variant in the medial prefrontal cortex, an AAV vector containing either variant (Cre-dependent) was mixed with a low-titer Cre-expressing AAV vector and injected into the medial prefrontal cortex as described in *Mahn et al., 2018*. The titers of purified AAV vectors used for injections were, in genome copies per milliliter (gc/ml): Cre, 5.6E9; stCoChR, 2.1E12; non-targeted CoChR, 7.2E11; soCoChR, 6.2E11.

## In vitro electrophysiological recordings and characterization of CoChR variants

Acute medial prefrontal cortex slices were prepared as described in *Mahn et al., 2018*. Whole-cell patch-clamp recordings were obtained under visual control using oblique illumination on a two-photon laser-scanning microscope (Ultima IV, Bruker, Billerica, MA) equipped with a femtosecond pulsed laser (Chameleon Vision II, 80 MHz repetition rate; Coherent, CA), a 12 bit monochrome CCD camera (QImaging QIClick-R-F-M-12) and a 20x, 1 NA objective (Olympus XLUMPlanFL N, Tokyo, Japan). Borosilicate glass pipettes (Sutter Instrument BF100-58-10, Novato, CA) with resistances ranging from 3 to 4 MΩ were pulled using a laser micropipette puller (Sutter Instrument Model P-2000). Recordings were obtained in carbogenated artificial cerebrospinal fluid ([mM] 3 KCl, 11 glucose, 123 NaCl, 26 NaHCO3, 1.25 NaH2PO4, 1 MgCl2, 2 CaCl2; 300 mOsm/kg) supplemented with the glutamate receptor blockers APV (25 µM) and CNQX (10 µM). The recording chamber was perfused at 2 ml/min and maintained at ~27°C. Pipettes were filled with Cs-based intracellular solution ([mM] 5 CsCl, 120 CsMeSO$_3$, 10 HEPES, 10 Na$_2$-Phosphocreatine, 4 ATP-Mg, 0.3 GTP-Na, 5 QX-314-Cl; 285 mOsm/kg; pH adjusted to 7.25 with CsOH) which contained Alexa fluor 350 dye (<1 mM, Thermo Fisher Scientific) and Neurobiotin Tracer (0.3 mg/ml, Vector Laboratories). Once a whole-cell recording from a prefrontal cell expressing one of the CoChR variants was obtained, peak photocurrent was measured under one-photon full-field illumination (illumination area ~0.66 mm$^2$). After the Alexa dye has diffused across the cell, a reference two-photon image stack was scanned (720 nm, 2.5 µm intervals in z axis) and the neurites of the recorded cell were manually traced in three dimensions. Multiple points (9–62) were manually selected along the cell's neurites and targeted for two-photon spiral stimulation (spiral diameter 10 µm, distance between revolutions 1 µm, 15 mW on sample, 10.053 ms duration). The stimulation was repeated twice, once in forward and once in reverse order. For soCoChR cells, the stimulation was repeated also using 40 mW on sample, except for one cell. Recordings were performed using a MultiClamp 700B amplifier, online-filtered at 10 kHz and digitized at 50 kHz using a Digidata 1440A digitizer (Molecular Devices, San Jose, CA).

## Viral injections for in vivo experiments

Viral injections were performed either on pups (P1 - P2, with P0 indicating the day of birth) or in young adults (>P28) similarly to *Brondi et al., 2020*; *Zucca et al., 2019*. Briefly, P1 – P2 injection of stCoChR-NLS:eGFP were performed on pups previously anaesthetized via hypothermia and immobilized on a refrigerated stereotaxic apparatus. A small incision on the skin was performed to expose the skull over one hemisphere and ~250 nl of viral solution was slowly injected through a glass pipette (1 mm lateral from bregma, at 0.25 mm depth). At the end of the injection, the skin was sutured and the pup revitalized under a heating lamp. Experiments were performed 4–12 weeks after the injection (*Figures 3–6*, *Figure 3—figure supplement 1*, ; *Figure 3—figure supplement 3A*; *Figure 3—figure supplement 3B*; ). For the experiments displayed in *Figure 2*, *Figure 3—figure supplement 3C*, *Figure 3—figure supplement 3D*, *Figure 4—figure supplement 1*, *Figure 5—figure supplement 1*, *Figure 5—figure supplement 3*, injections were performed on 4–10 weeks old animals. Once young adults animals were anesthetized with 2% isoflurane/0.8% oxygen, they were placed in a stereotaxic apparatus (Stoelting Co, Wood Dale, IL), while the temperature was maintained at 37°C with a heating pad. The head was shaved and disinfected, and a small incision was performed to expose the skull over the primary somatosensory cortex. One to three small holes were drilled on the skull in order to lower a glass micropipette containing the viral solution into the parenchyma (pipette depth: 0.2–0.3 mm from brain surface). 300 nL of virus *per* site were injected at 30–50 nL/min by means of a hydraulic injection apparatus driven by a syringe pump (UltraMicro-Pump, WPI, Sarasota, FL). At the end of the injection, the scalp incision was sutured, covered with antibiotic ointment and the animals were placed under a heating lamp until full recovery. Experiments were performed 3–16 weeks after injection. For a subset of crosstalk experiments (*Figure 5D–F*) and for simultaneous imaging and photo-stimulation experiments (*Figure 6*, *Figure 5—figure supplement 2*, *Figure 5—figure supplement 3*, *Figure 6—figure supplement 1*, *Figure 6—figure supplement 2*), injection of the AAV transducing the opsin and the AAV carrying the jRCaMP1a construct were performed in two subsequent injections with procedures similar to those described previously.

## Animal surgery

Surgical procedures prior to electrophysiological recordings, photo-stimulation, and imaging experiments were performed on urethane anesthetized young-adult animals (3–16 weeks old, either sex), as described previously (*Beltramo et al., 2013*; *Vecchia et al., 2020*; *Zucca et al., 2017*). Mice were anesthetized with an intraperitoneal injection of urethane (16.5%, 1.65 g/kg). Animal body temperature was kept constant at 37°C with a heating pad and monitored together with respiration rate, vibrissae movement, and reactions to tail pinching throughout the surgery and the experiments. The scalp was removed after infiltrating all incisions with lidocaine. Head-fixation of mice was achieved using custom printed plastic chambers with a 4 mm central hole, which was attached to the animal's skull by means of superglue and dental cement. A craniotomy (area: ~700×700 mm$^2$) was opened over the right sensory cortex close to the injection sites (identified by looking at the fluorescence of the expressed transgene) and the dura was carefully removed. The surface of the brain was kept moist with standard HEPES-buffered artificial cerebrospinal fluid (aCSF) composed of (in mM): 127 NaCl, 3.2 KCl, 2 CaCl$_2$, and 10 HEPES at pH 7.4.

## Optical setup for in vivo recordings

The optical set-up for two-photon holographic illumination at high repetition rate (80 MHz, *Figures 2* and *3* and *Figure 3—figure supplement 2*) was composed of an ultrafast pulsed laser source (S1 in *Figure 2* and *Figure 6*, Chameleon Discovery, 80 MHz repetition rate, tuned at 920 nm or 1100 nm, Coherent, Milan, IT), a customized scan-head (Bruker Corporation, former Prairie Technologies, Milan, IT), an upright epifluorescence microscope (BX61, Olympus, Milan, IT), and a liquid crystal spatial light modulator (SLM, X10468-07 SLM, Hamamatsu, Milan, IT), which was conjugated to the back aperture of the objective (*Dal Maschio et al., 2010*; *Dal Maschio et al., 2011*). The laser beam intensity was modulated by a Pockels cell (P$_1$ in *Figures 2* and *3*, Conoptics Inc, Danbury, CT) and then directed to the SLM by a sequence of mirrors (UM10AG, Thorlabs, Newton, NJ). A half-wave plate (RAC 5.2.10 achromatic λ/2 retarder - B. Halle Nachfl GMBH, Berlin, DE) was placed before the SLM in order to obtain the optimal polarization for phase-only modulation at the SLM. A first telescope (IR doublets 30 mm and 75 mm, Thorlabs, Newton, NJ) expanded the laser beam to fill the active window of the SLM. A second telescope (IR doublets 300 mm and 150 mm, Thorlabs, Newton, NJ) was used to resize the laser beam to fit the dimensions of the scanning mirrors inside the scan-head (G in *Figure 2*, G2 in *Figure 6*) and to optically conjugate the plane of the SLM with the back aperture of the objective. Two multi-alkali photomultipliers tubes (PMTs, Hamamatsu, Milan, IT) were used as detectors for raster scanning imaging. Dual emission filters in front of the two PMTs were 525/70 nm and 607/45 nm, respectively. In *Figure 2* and *Figure 6*, D$_1$ was a 660 nm long-pass dichroic mirror, D$_2$ a 575 nm long-pass dichroic mirror; in *Figure 6*, D$_3$ was a 980 nm long-pass dichroic mirror. The Nikon CFI75 LWD 16X W (0.8 NA, Nikon, Tokyo, Japan) was used for most experiments, except those in *Figure 2*, *Figure 3—figure supplement 2*, *Figure 3—figure supplement 3*, *Figure 4—figure supplement 1*, *Figure 5—figure supplement 1*, in which the Olympus LUMPlanFl40X/IR objective (0.8 NA, Olympus, Tokyo, Japan) was used. Holographic illumination was controlled by an analog pulse of variable amplitude (0–1.6 V), generated by the Digidata 1440 (Axon instruments, Union City, CA) and conveyed into the Pockels cell amplifier.

Holographic illumination at low repetition rate (1 MHz or below, *Figures 3*, *4* and *6*, *Figure 3—figure supplement 1*, *Figure 3—figure supplement 2*, *Figure 3—figure supplement 3*, *Figure 4—figure supplement 1*, *Figure 6—figure supplement 1*, *Figure 6—figure supplement 2*) was achieved by adding a second laser source to the optical path. The source was composed of a high energy femtosecond laser (Monaco 40 W; Coherent, Milano, IT or Carbide 45 W: Light conversion, Vilnius, Lithuania) combined with an optical parametric amplifier (OPA) (Opera-F; Coherent, Milano, IT or Orpheus-F; Light conversion, Vilnius, Lithuania, respectively). The signal output wavelength of the OPA was set at 920 nm. Repetition rate was 1 MHz or below (via AOM pulse-picking). A pulse compressor after the OPA was used in some experiments. The pulse length after the compressor was <290 fs. Laser source for photostimulation could be switched by using a motorized mirror in the optical path.

For simultaneous two-photon imaging and stimulation experiments (*Figure 6*, *Figure 6—figure supplement 1*, *Figure 6—figure supplement 2*) a low repetition rate high energy laser source (Carbide + Orpheus) was used for holographic illumination and tuned at 920 nm. The telescope

downstream the SLM was replaced by two IR doublets (400 mm and 125 mm, Thorlabs, Newton, NJ) and the stimulation beam was relayed onto a second set of galvanometric mirrors inside the scan-head (mirror diameter: 3 mm). Imaging and stimulation beams were combined by a dichroic mirror (zt980rdc, Chroma Technology Corporation, Bellows Falls, VT) positioned between the two sets of galvanometric mirrors and the scan lens. Phase modulation for holographic illumination and calibration of the optical setup were done as previously described (*Forli et al., 2018*).

To control for the functionality of excitatory opsins in the recorded neurons, in a subset of the experiments single-photon stimulation of opsins was performed at 488 nm using a laser (MLD, COBOLT, Solna, SE) and a multimode fiber to deliver light to the brain (core diameter 200 µm, 0.22 NA, QMMJ-3X-UVVIS-200/240–0.4-6, OZ Optics Ldt, Ottawa, CA). The laser was coupled to the fiber via a 10X objective (MPLN10X, Olympus, Milan, IT). On-off control of illumination was performed directly with a TTL input to the laser driver (stimulus duration: 50 ms). Light intensity was 150–300 µW at the fiber tip. The optical fiber was positioned ~500 µm above the craniotomy, at an angle of ~30°.

## In vivo electrophysiological recordings

We performed two-photon targeted juxtasomal electrophysiological recordings as previously described in *Bovetti et al., 2017*; *De Stasi et al., 2016*. The Sutter P-97 micropipette puller (Sutter instrument, Novato, CA) was used to pull Borosilicate glass pipettes (Hilgenberg, Malsfeld, Germany) with a resistance of 4–9 MΩ. Pipettes were filled with aCSF solution mixed with Alexa Fluor 488 or 594 (10 mM, Thermo Fisher Scientific, Waltham, MA). stCoChR-expressing neurons in L2/3 were targeted under the two-photon microscope by imaging the expressed fluorescent reporter (eGFP or mScarlet, at 920 nm and 1050 nm excitation wavelength, respectively) while monitoring the pipette fluorescence and its electrical resistance (through brief voltage steps). When the pipette tip and the target cell were in close contact one to the other, a patch of cell membrane was sealed on the pipette tip by applying a mild negative pressure. The juxtasomal configuration was reached when the pipette resistance was >20 MΩ and spikes were clearly visible. Extracellular AP waveforms from the target neuron were recorded in current-clamp during spontaneous activity and holographic illumination epochs. Electrical signals were amplified by a Multiclamp 700B, low-pass filtered at 2.2 kHz, digitized at 50 kHz with a Digidata 1440, and acquired with pClamp 10 (Molecular Device, Sunnyvale, CA). Analysis of electrophysiological recordings was carried out using Clampfit 10.4 software (Molecular Device, San Jose, CA), IgorPro (WaveMetrics, Portland, OR) and OriginPro 2018 (Origin-Lab, Northampton, MA). In vivo electrophysiological recordings for crosstalk quantification (*Figure 5* and *Figure 5—figure supplement 1*) were performed on stCoChR-expressing neurons while raster scanning the recorded cell at 11 Hz (pixel size = 0.6–1.6 µm) at both 1100 nm and 920 nm excitation wavelengths (average power:≤35 mW).

## All-optical two-photon imaging and holographic stimulation in vivo

Simultaneous two-photon imaging and photostimulation experiments were performed in anesthetized mice expressing jRCaMP1a and stCoChR in cortical neurons. Imaged field of views were located in L2/3 of the primary somatosensory cortex (average depth of recordings: 145 µm). Two-photon imaging at $\lambda_{exc}$ = 1050 nm was performed to assess the expression pattern of the opsin (stCoChR) and of the calcium indicator (jRCaMP1a) at the same time. A reference image of the selected FOV was acquired, and holographic oval shapes covering the soma of target neurons were generated by the SLM ($\lambda_{exc}$ = 920 nm) and projected at the sample. Temporal series were acquired in raster scanning configuration with the imaging beam (image dimension: 100 × 100 pixels; frame rate: 11 Hz; pixel dwell time: 4 µs; $\lambda_{exc}$ = 1100 nm). Holographic photostimulation duration was 200 ms and it was repeated six times at 0.066 Hz.

## Immunohistochemistry for subcellular localization of CoChR and stCoChR

Brains of mice sparsely expressing either CoChR or stCoChR in the medial prefrontal cortex were fixated and sectioned as described in *Mahn et al., 2018*. Coronal cortical sections (50 µm thick) were washed three times in phosphate-buffered saline (PBS) and permeabilized in 0.5% Triton (Sigma-Aldrich, Rehovot, Israel) in PBS for 1 hr at room temperature, followed by incubation in blocking

solution (20% normal horse serum [NHS] and 0.3% Triton in PBS) for 1 hr at room temperature. Sections were then exposed to polyclonal rabbit anti-2A peptide primary antibody (diluted 1:500 in PBS with 5% NHS and 2% Triton; Millipore Cat# ABS31, RRID:AB_11214282) for 48 hr at 4℃. Following three washes in PBS, sections were exposed to polyclonal Cy5-conjugated donkey anti-rabbit secondary antibody (diluted 1:500 in PBS with 2% NHS; Jackson ImmunoResearch Labs Cat# 711-175-152, RRID:AB_2340607) for 2 hr at room temperature. Sections were washed three times in PBS, incubated with DAPI (5 mg/ml solution diluted 1:30,000 prior to staining; Thermo Fisher Scientific) for 5 min at room temperature, washed again for three times in PBS, and embedded in DABCO mounting medium (Sigma-Aldrich) on gelatin-coated slides. Slides were imaged using a confocal microscope (Zeiss LSM 700 or Leica TCS SP5) under identical conditions. Images of cells expressing CoChR or stCoChR were analyzed using Fiji software.

## Data analysis

For the calculation of the decay length constants ($\tau_{decay}$) as a measure for soma restriction in vitro (**Figure 1E**), the distance of each stimulated point from the soma along the path of the neurites was measured by extending a segmented line from the soma, along the neurites to the relevant point on the lateral plane using Fiji software (version 1.52) (**Schindelin et al., 2012**). Distance along the axial axis was neglected. Peak photocurrent at each point was calculated and normalized to the photocurrent at the soma. $\tau_{decay}$ was calculated for each cell using a monoexponential fit.

Electrophysiological traces from in vivo juxtasomal recordings were filtered with a high-pass filter with a cutoff frequency of 10 Hz and spikes were detected using a threshold criterion. The threshold was visually adjusted for each sweeps by an expert user and it was set at >3 the standard deviation of the noise. For experiments in **Figures 2** and **3**, **Figure 3—figure supplement 2**, **Figure 6—figure supplement 1**, traces were divided in three time windows: before (Pre, 1 s), during (Stim, 0.1 s), and after (Post, 1.5 s) holographic illumination. $\Delta_{AP}$Freq was defined as the difference between the average firing frequencies in the Stim and Pre windows. The spatial resolution was calculated as follows: neuronal responses to photostimulation ($\Delta_{AP}$Freq) were recorded as a function of the distance between the soma of the targeted neuron and the projected holographic shape (**Figure 2I and J**), which was radially (radial shift: 0, 20, 40, 60 μm) or axially (axial shift: −75,–50, −25, 0, 25, 50, 75 μm) shifted. $\Delta_{AP}$Freq as a function of the shift in three different directions (radial, axial$_{up}$, and axial$_{down}$) was fitted with a mono-exponential function:

$$\Delta APFreq(x) = A * e^{(-b*x)}$$

as described in **Packer et al., 2015**. We excluded curves fittings with b < 0 or with values of A showing a difference >25% compared to the $\Delta_{AP}$Freq value computed when the shape was centered with the target neuron (radial shift: 0 μm). The spatial resolution, l$_{1/2}$, was defined as the distance at which half of the evoked response calculated from fit (A/2) was observed. Axial$_{up}$ was defined as the direction toward the brain surface, while axial$_{down}$ was defined as the ventral direction. For **Figures 2–4** and **Figure 3—figure supplement 1**, **Figure 3—figure supplement 2**, **Figure 4—figure supplement 1**, AP probability (AP Prob.) was defined as the number of stimulation trials in which one or more APs were recorded during the illumination period (or shorter epochs) divided by the total number of stimulation trials.

For **Figure 3—figure supplement 3**, data were fitted with a decreasing exponential function:

$$y = y_0 + Ae^{-\frac{x}{t}}$$

where y was the latency/jitter, x the average power, and parameters were not fixed, except for $y_0$. In panels **Figure 3—figure supplement 3A and C**, $y_0$ was set according to the average latency observed with 1P stimulation. In panels **Figure 3—figure supplement 3B and D**, $y_0$ was set to zero.

For all-optical experiments in **Figure 6**, temporal series acquired in vivo were analyzed using custom scripts written in MATLAB (Mathworks, Natick, MA). Stimulated regions of interest (ROIs) were drawn and, for each ROI, the change in fluorescence relative to the baseline ($\Delta F/F_0$) was computed as a function of time for jRCaMP1a signals. The fluorescence jRCaMP1a baseline ($F_0$) was calculated in ten frames at the beginning of the recorded session. When above noise level, fluorescence artifacts due to holographic stimulation were removed by blanking the jRCaMP1a signal in the frames corresponding to stimulation periods. Fluorescent jRCaMP1a transients were fitted with a mono-

exponential function and the corresponding amplitude of the transient at the offset of stimulation and decay time were calculated. Repetition rate during holographic stimulation was 1 MHz, 500 kHz, and 100 kHz. A cell was defined 'responsive' to the holographic stimulation if the amplitude of the average (across stimulation trials) fluorescence transient at the offset of holographic illumination was larger than three times the standard deviation of the trace measured during a pre-stimulation period. The success rate of all-optical manipulation at 1 MHz repetition rate was 85% (28 responsive neurons out of 33 stimulated neurons). Responsive cells decreased with the laser repetition rate. At 0.5 MHz the percentage of responsive cells was 73% (24 responsive neurons out of 33 stimulated neurons); at 0.1 MHz the percentage of responsive cells was 39% (13 responsive neurons out of 33 stimulated neurons). In responsive neurons stimulated at 1 MHz repetition rate, the average $\Delta F/F_0$ ratio of stimulated calcium transients was 23%±2, N = 28 cells from two mice. For all-optical experiments and simultaneous juxtasomal recordings (*Figure 6F,G*, *Figure 5—figure supplement 2*, *Figure 6—figure supplement 1*, *Figure 6—figure supplement 2*), the amplitude of fluorescence transients corresponding to AP events was calculated as described previously (*Forli et al., 2018*). Decay time for spontaneous jRCaMP1a activity (*Figure 6—figure supplement 2D*) was calculated for each recorded neuron by fitting $\Delta F/F_0$ traces with an autoregressive process of order 2 (*Pnevmatikakis et al., 2016*).

## Statistics

All values were expressed as mean ± SEM, unless otherwise stated. Sample size (n) for different experiments was chosen based on previous studies (*Forli et al., 2018*; *Packer et al., 2015*; *Yang et al., 2018*). Blinding was not used in this study. Analysis presented in this manuscript included all recordings with no technical issues. For n $\geq$ 10, a Kolmogorov-Smirnov test was used while, for n < 10, a Saphiro-Wilk test or a Kolmogorov-Smirnov test (n < 5) were adopted to test for normality. Student's *t*-test was used to calculate statistical significance when comparing two populations of normally distributed data. The non-parametric Mann-Whitney U test or Wilcoxon signed-rank test (for unpaired or paired comparison, respectively) was used in case of non-normal distributions, unless otherwise stated. One-way ANOVA with Bonferroni or Tukey's post hoc test was used when multiple (>2) normally distributed populations of data were compared. For non-normal distribution and multiple comparisons, the non-parametric Friedman test with Dunn's post hoc correction and the Kruskal-Wallis test with Tukey's post hoc HSD were used. All tests were two-sided. Statistical analysis was performed using Prism 6 (GraphPad, La Jolla, CA), OriginPro 2018 (OriginLab, Northampton, MA), and MATLAB (Mathworks, Natick, MA).

## Acknowledgements

We thank D Vecchia for comments on the manuscript, K Deisseroth for opsin plasmids, and V Jayaraman, DS Kim, LL Looger, K Svoboda, from the GENIE Project, Janelia Research Campus, Howard Hughes Medical Institute for jRCaMP1a expressing AAVs. This work was supported by ERC (NEURO-PATTERNS 647725), NIH (U01 NS090576, U19 NS107464), H2020-RIA (DEEPER 101016787), and by the ERC (PrefrontalMap 819496), the Human Frontier Science Program, the Brain and Behavior Research Foundation, the Ilse Katz Institute for Material Sciences and Magnetic Resonance Research, the Adelis Prize for Brain Research and the Candice Appleton Family Trust.

## Additional information

### Funding

| Funder | Grant reference number | Author |
|---|---|---|
| H2020 European Research Council | 647725 | Tommaso Fellin |
| NIH | U01 NS090576 | Tommaso Fellin |
| NIH | U19 NS107464 | Tommaso Fellin |
| H2020 European Research Council | DEEPER 101016787 | Ofer Yizhar Tommaso Fellin |

| H2020 European Research Council | 819496 | Ofer Yizhar |
|---|---|---|
| Human Frontier Science Program | RGY0064/2017 | Ofer Yizhar |
| Brain and Behavior Research Foundation | | Ofer Yizhar |
| Ilse Katz Institute for Material Sciences and Magnetic Resonance Research | | Ofer Yizhar |
| Adelis Brain Research Award | | Ofer Yizhar |

The funders had no role in study design, data collection and interpretation, or the decision to submit the work for publication.

### Author contributions

Angelo Forli, Conceptualization, Data curation, Software, Formal analysis, Validation, Investigation, Visualization, Methodology, Writing - original draft, Writing - review and editing; Matteo Pisoni, Data curation, Formal analysis, Validation, Investigation, Visualization, Methodology, Writing - original draft, Writing - review and editing; Yoav Printz, Conceptualization, Data curation, Formal analysis, Validation, Investigation, Visualization, Methodology, Writing - original draft, Writing - review and editing; Ofer Yizhar, Tommaso Fellin, Conceptualization, Resources, Supervision, Funding acquisition, Writing - original draft, Project administration, Writing - review and editing

### Author ORCIDs

Matteo Pisoni  https://orcid.org/0000-0003-0480-5220
Yoav Printz  https://orcid.org/0000-0002-5071-5640
Ofer Yizhar  https://orcid.org/0000-0003-4228-1448
Tommaso Fellin  https://orcid.org/0000-0003-2718-7533

### Ethics

Animal experimentation: All experiments involving animals were approved by the IIT Animal Welfare Body, by the National Council on Animal Care of the Italian Ministry of Health (authorization #34/2015-PR, #1084/2020-PR), and by the Institutional Animal Care and Use Committee at the Weizmann Institute of Science, and carried out in accordance with the guidelines established by the European Communities Council Directive.

### Decision letter and Author response

Decision letter https://doi.org/10.7554/eLife.63359.sa1
Author response https://doi.org/10.7554/eLife.63359.sa2

## Additional files

### Supplementary files

• Source code 1. Source code for the analysis of spontaneous jRCaMP1a transients.

• Source code 2. Source code for the analysis of all-optical recordings.

• Source code 3. Source code for the extraction of fluorescence traces.

• Source code 4. Source code for the extraction of fluorescence traces.

• Supplementary file 1. Supplementary tables comparing photocurrent properties of CoChR, soCoChR, and stCoChR. Supplementary Table 1.Photocurrents elicited by two-photon spiral scan over the soma for CoChR variants. 'v' marks under each post hoc p value denote which CoChR variants were compared post hoc to obtain each specific p value. This also applies to Supplementary Table 2–4. # indicates that a separate Kruskal-Wallis test with separate post hoc comparisons was performed for comparing soCoChR under 40 mW with stCoChR and CoChR, since the two light powers used for soCoChR are paired. The row of soCoChR @ 40 mW presents the

results of this separate set of tests, in which soCoChR @ 40 mW replaces soCoChR in the post hoc comparisons. This applies also to Supplementary Table 3–4. Supplementary Table 2. Photocurrents elicited by one-photon full-field illumination of CoChR variants. Supplementary Table 3. Soma-to-full-field photocurrent ratio of CoChR variants. Supplementary Table 4. τdecay of photocurrent with distance from soma along neurites for CoChR variants.

- Transparent reporting form

### Data availability

We provide Source Data for the data plotted in all figures as Excel Source Data files.

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
