## [Decision Letter]

**Acceptance summary:**

In this manuscript, the authors developed a soma-targeted variant of an existing blue-light sensitive opsin for efficient optogenetic stimulation. It allows neuronal stimulation with ten-fold lower power and low spectral crosstalk with red calcium indicator. This newly developed opsin presents a new powerful tool for a large-scale all-optical control of neuronal activity in vivo.

**Decision letter after peer review:**

Thank you for submitting your article "Optogenetic strategies for high-efficiency crosstalk-free all-optical interrogation using blue light-sensitive opsins" for consideration by *eLife*. Your article has been reviewed by 3 peer reviewers, and the evaluation has been overseen by a Reviewing Editor and John Huguenard as the Senior Editor. The following individual involved in review of your submission has agreed to reveal their identity: Shai Berlin (Reviewer #1).

The reviewers have discussed the reviews with one another and the Reviewing Editor has drafted this decision to help you prepare a revised submission.

Summary:

In this manuscript, the authors developed a soma-targeted variant of an existing blue-light sensitive opsin, CoChR, for efficient optogenetic stimulation and combined it with red calcium indicator aiming to achieve low crosstalk in all-optical control of neuronal activity. Compared to a previous soma-restricted version of CoChR, the new stCoChR achieves soma restriction without compromising photocurrent, thus enabling efficient two-photon neuronal excitation in vivo. The optimal laser power and repetition rate for driving spikes in stCoChR positive neurons were carefully calibrated with juxtasomal recordings in vivo. The authors tested the crosstalk between two-photon raster scanning at a wavelength typically used for imaging red functional indicators and photoactivation of stCoChR-expressing neurons. They also demonstrated simultaneous two-photon imaging and holographic photostimulation of L2/3 neurons in mice.

While the reviewers are impressed by the efficiency of the new opsin variant, which will be useful for the growing community of experimenters performing "all-optical experiments", the claim of 'crosstalk-free' all-optical interrogation – stated in the title! – is not fully convincing. Therefore, the advance presented here is being oversold, unless the authors provide further evidence as detailed below.

Essential revisions:

1. The experiments in Figure 5 and Figure S5 aim to quantify the crosstalk between imaging and photostimulation, which is the key advantage of the proposed method over previous all-optical approaches. However, these experiments were performed in neurons expressing stCoChR without RCaMP.

The concern is whether the imaging conditions used here can provide sufficient RCaMP signal, if RCaMP is co-expressed. Imaging quality is particularly important here given that jRCaMP1a is not as bright as GCaMP6s as adopted in many other all-optical studies. The authors argued they used 'imaging conditions commonly used to monitor red-shifted functional indicators', but the expression of stCoChR may affect the expression of RCaMP, and therefore requires more power for imaging the same neurons.

Ideally these cross-talk experiments should be done in neurons co-expressing stCoChR and RCaMP, with simultaneous imaging and juxtasomal recording to confirm that the imaging conditions used here is sufficient to report spikes in the jRCaMP1 signal while insufficient to activate the opsin.

2. What is the success rate of all-optical manipulation? Specifically, what is the percentage of targeted cells that showed fluorescence transients with adequate signal-to-noise? How does this percentage vary with different photostimulation laser repetition rates and power? Are the traces in Figure 6C showing all targeted neurons in one experiment?

3. Will increasing imaging power assist the detection of photostimulation-evoked RCaMP events? As one increases the imaging laser power, at what point does it stop improving detection of RCaMP events immediately following photostimulation, and at what point does it start to increase the baseline RCaMP events in the opsin-positive neurons in the field-of-view? Answers to these questions will also assist the evaluation of crosstalk between RCaMP imaging and stCoChR activation.

4. The authors show that reducing the repetition rate of the laser allows holographic activation of stCoChR at low average power. A biophysical explanation for this observation is not given. The results are clearly presented and the new tool is certainly of interest for in-vivo optogenetics labs, as there is no detectable interference between calcium imaging and photostimulation. A caveat is the relatively poor control of spike timing: The method requires 20 ms of 2p excitation to reliably generate an action potential, while blue light pulses of 1 ms are usually sufficient. Given that the closing of stCoChR is also relatively slow, it would be of interest to know the maximum frequency at which single spike control is possible with this approach. Also, the z-resolution is poor (>80 µm), which could be a problem for dense cell layers. In a methods paper, these limitations should be discussed.

5. Because of the quadratic dependence of excitation events on pulse intensity, 2p imaging/photoactivation at lower pulse repetition rates can be performed at lower average power – this is not a miracle. Whether a low rep rate/high pulse energy regime confers any real advantage depends on the nature of the expected photodamage: If thermal effects dominate, low average power is a good thing. The advantage of reduced repetition rates (and very short pulses!) for two-photon imaging of retinal has recently been published (Palczewska et al., PNAS, https://doi.org/10.1073/pnas.2007527117), this paper should be cited. If, on the other hand, triplet states and solvated electrons are produced and lead to the destruction of biomolecules, high peak powers are to be avoided. For 2p imaging, it has been shown that higher rep rates (with lower pulse power) strongly reduce the amount of photobleaching (Fuzeng Niu et al. 2019 Laser Phys. 29 046001). Will ever higher peak powers at some point be a problem for CoChR, do they bleach the retinal? A discussion along these lines would leave the reader less mystified (as the authors do not show a lower limit/optimum for the rep rate).

6. The authors do not provide a single micrograph showing the localization of stCoChR. As this is one of their main contributions here, they should provide high resolution micrographs showing the localization of the stCoChR in neurons. This could be done by immunolabelling or by simply removing the P2A sequence, and trying to see whether these are really soma targeted (and to what extent in comparison to dendrites, for instance).

7. What is missing throughout the paper are control experiments showing that without opsins, these different illumination schemes do not evoke firing. This is really missing in the all-optical interrogation of neurons in Figure 6. The authors need to show that under identical imaging settings, neurons that do not express stCoChR (but that positively express jRCaMP1a) do not fire! This experiment (Figure 6) also requires recording of electrical activity along the optical responses and this is because in the example they show (Figure 6C), the responses do not look like APs, rather small increases in fluorescence due to low Ca^2+^ elevations. The authors should also show longer recordings (as they do in Figure 5B for instance) and then compare between the optical responses (DF/F) of jRCAMP to spontaneous APs versus those obtained by the optical stimulation of stCoChR. Are they different (DF/F, kinetics, etc).?

---

## [Author Response]

Essential revisions:1. The experiments in Figure 5 and Figure S5 aim to quantify the crosstalk between imaging and photostimulation, which is the key advantage of the proposed method over previous all-optical approaches. However, these experiments were performed in neurons expressing stCoChR without RCaMP.The concern is whether the imaging conditions used here can provide sufficient RCaMP signal, if RCaMP is co-expressed. Imaging quality is particularly important here given that jRCaMP1a is not as bright as GCaMP6s as adopted in many other all-optical studies. The authors argued they used 'imaging conditions commonly used to monitor red-shifted functional indicators', but the expression of stCoChR may affect the expression of RCaMP, and therefore requires more power for imaging the same neurons.Ideally these cross-talk experiments should be done in neurons co-expressing stCoChR and RCaMP, with simultaneous imaging and juxtasomal recording to confirm that the imaging conditions used here is sufficient to report spikes in the jRCaMP1 signal while insufficient to activate the opsin.

As requested by the referees, we repeated the crosstalk experiment in vivo in individual neurons co-expressing stCoChR and jRCaMP1a while simultaneously performing functional imaging and juxtasomal electrophysiological recording (new Figure 5D-F). Similarly to what previously observed in cells expressing only stCoChR, we found that raster scanning at 1100 nm (imaging power: 30-35 mW) did not alter the spontaneous firing rate of imaged neurons (new Figure 5E-F). As expected raster scanning at 920 nm, the wavelength of efficient two-photon excitation of stCoChR, caused a large and significant increase in the neuron’s spontaneous firing rate (new Figure 5E-F). Importantly, using imaging power between 30 mW and 35 mW allowed the detection of jRCaMP1a fluorescence transients (Figure 5G). Detectable jRCaMP1a events were mainly associated with the discharge of multiple action potentials similarly to previous reports (Dana et al. 2016, Forli et al. 2018). These experiments are now described in the Results section at lines 254-263. Following referees’ concern about jRCaMP1a performance, we modified the Discussion (lines 397-407) to highlight the limitations of current red-shifted sensors and the importance of future improvements.

2. What is the success rate of all-optical manipulation? Specifically, what is the percentage of targeted cells that showed fluorescence transients with adequate signal-to-noise? How does this percentage vary with different photostimulation laser repetition rates and power? Are the traces in Figure 6C showing all targeted neurons in one experiment?

The success rate of all-optical manipulation was calculated at 1 MHz repetition rate at was 85 % (28 responsive neurons out of 33 stimulated neurons). This information is now provided at lines 891-892.

In responsive neurons, the average ΔF/F_0_ ratio of stimulated calcium transients was 23 ± 2 % (n = 28 cells from 2 mice), well above the noise level. This information is now provided on line 895-897.

The percentage of responsive cells decreased with the laser repetition rate: at 0.5 MHz, the percentage of responsive cells was 73 % (24 responsive neurons out of 33 stimulated neurons); at 0.1 MHz, the percentage of responsive cells was 39 % (13 responsive neurons out of 33 stimulated neurons). This information is now provided on line 892-895. We did not quantitatively explore how responsive cell percentage varied with illumination average power.

Traces in figure 6C show all the 6 neurons targeted in that example experiment. This information is now provided on line 600.

3. Will increasing imaging power assist the detection of photostimulation-evoked RCaMP events? As one increases the imaging laser power, at what point does it stop improving detection of RCaMP events immediately following photostimulation, and at what point does it start to increase the baseline RCaMP events in the opsin-positive neurons in the field-of-view? Answers to these questions will also assist the evaluation of crosstalk between RCaMP imaging and stCoChR activation.

To address the referees’ questions, we first compared the results obtained performing jRCaMP1a imaging at 30-35 mW and at 50 mW average illumination power. In combined imaging and juxtasomal electrophysiological recordings, we found that the ∆F/F_0_ of jRCaMP1a transients tended to be higher at 50 mW but the effect was not statistically significant (new Figure 5—figure supplement 2). We then evaluated crosstalk between imaging and photostimulation at the higher imaging average power (i.e., 50 mW). We found that raster scanning induced a small, but significant, increase in spontaneous firing frequency (new Figure 5—figure supplement 3). This was true in neurons expressing stCoChR (Figure 5—figure supplement 3 left) and in neurons co-expressing stCoChR and jRCaMP1a (Figure 5—figure supplement 3 right). Thus, on the one hand increasing imaging power did not significantly improve the ∆F/F_0_ of jRCaMP1a transients. On the other hand, imaging at higher average power started causing small but significant crosstalk. These results are now reported at lines 268-272.

Given that with these new experiments we evaluated the threshold for inducing crosstalk under our experimental conditions, we modified the title of the paper. It now reads: “Optogenetic strategies for high-efficiency all-optical interrogation using blue light-sensitive opsins”. We thank the referees for their suggestions and constructive comments on this aspect of our study.

4. The authors show that reducing the repetition rate of the laser allows holographic activation of stCoChR at low average power. A biophysical explanation for this observation is not given. The results are clearly presented and the new tool is certainly of interest for in-vivo optogenetics labs, as there is no detectable interference between calcium imaging and photostimulation. A caveat is the relatively poor control of spike timing: The method requires 20 ms of 2p excitation to reliably generate an action potential, while blue light pulses of 1 ms are usually sufficient. Given that the closing of stCoChR is also relatively slow, it would be of interest to know the maximum frequency at which single spike control is possible with this approach. Also, the z-resolution is poor (>80 µm), which could be a problem for dense cell layers. In a methods paper, these limitations should be discussed.

Holographic activation of stCoChR at low average power is achieved by reducing the repetition rate – as pointed out by the referees – but also by using higher energy *per* pulse. The overall effect can be explained by considering the relationship between the probability of a two-photon absorption process, the power of the stimulating laser, and the repetition rate of the laser source (Denk and Webb, 1990; Zipfel at al., 2003).

To address the request of the referees regarding single spike control, we performed new juxtasomal electrophysiological recordings in cell expressing stCoChR during stimulation with trains of short pulses (pulse duration, 10 ms; number of pulses, 5; average power, 5 mW) at different frequency (20 Hz -50 Hz, new Figure 3—figure supplement 1). We found that the probability of eliciting at least one action potential raised from 20 Hz to 40 Hz and then tended to decrease at 50 Hz (Figure 3—figure supplement 1B). These results are described at lines 194-203.

Following the referees’ comment, we also added a paragraph to discuss the issue of z-resolution in the context of dense cellular staining and how the z-resolution could be improved. It reads:

“The soma-restriction method described in this study effectively increased the spatial resolution of photo-stimulation (Figure 1D, E). […] Spatial resolution could also be further improved by developing more effective soma-restriction motifs”.

5. Because of the quadratic dependence of excitation events on pulse intensity, 2p imaging/photoactivation at lower pulse repetition rates can be performed at lower average power – this is not a miracle. Whether a low rep rate/high pulse energy regime confers any real advantage depends on the nature of the expected photodamage: If thermal effects dominate, low average power is a good thing. The advantage of reduced repetition rates (and very short pulses!) for two-photon imaging of retinal has recently been published (Palczewska et al., PNAS, https://doi.org/10.1073/pnas.2007527117), this paper should be cited. If, on the other hand, triplet states and solvated electrons are produced and lead to the destruction of biomolecules, high peak powers are to be avoided. For 2p imaging, it has been shown that higher rep rates (with lower pulse power) strongly reduce the amount of photobleaching (Fuzeng Niu et al. 2019 Laser Phys. 29 046001). Will ever higher peak powers at some point be a problem for CoChR, do they bleach the retinal? A discussion along these lines would leave the reader less mystified (as the authors do not show a lower limit/optimum for the rep rate).

We thank the referees’ for their comment. Previous work shows that under the illumination conditions that we are using in our study (energy per pulse, ≤ 7 nJ), thermal effects dominate over non-linear photodamage effects (Picot et al. 2018, Charan et al. 2018). Moreover, the retinal group is the same in all opsins and previous work showed that energies up to 60-200 nJ per pulse can be used to activate opsins without causing retinal bleaching and cell damage (Mardinly et al. 2018, Gill et al. 2020, Chen et al. 2019). Thus, we expect that the peak energy could be further increased compared to the one used in our study, while preserving opsin functionality. We added a paragraph to the discussion on page lines 377-385 to clarify these points.

We thank the referees for pointing our attention to Palczewska et al. 2020, which is now cited in the manuscript at line 381.

6. The authors do not provide a single micrograph showing the localization of stCoChR. As this is one of their main contributions here, they should provide high resolution micrographs showing the localization of the stCoChR in neurons. This could be done by immunolabelling or by simply removing the P2A sequence, and trying to see whether these are really soma targeted (and to what extent in comparison to dendrites, for instance).

Following the referees’ request, we now provide confocal images showing CoChR and stCoChR expression in neurons (new Figure 1—figure supplement 1). However, we think that confocal image analysis of protein localization is not conclusive of a somatic localization due to the limited quantitative accuracy of immunostaining. Moreover, removing the P2A sequence, as the referees proposed, would change the sequence of stCoChR and might in itself modify the targeting efficiency of the CoChR-fluorophore fusion molecules. In contrast, we believe the functional measurements displayed in Figure 1 and Figure 1—figure supplement 2, in which we directly measured photocurrent induced by localized two-photon excitation as a function of the distance from the soma, as well as the soma-to-full-field photocurrent ratio, provide clear and conclusive evidence of the somatic restriction of stCoChR.

7. What is missing throughout the paper are control experiments showing that without opsins, these different illumination schemes do not evoke firing. This is really missing in the all-optical interrogation of neurons in Figure 6. The authors need to show that under identical imaging settings, neurons that do not express stCoChR (but that positively express jRCaMP1a) do not fire! This experiment (Figure 6) also requires recording of electrical activity along the optical responses and this is because in the example they show (Figure 6C), the responses do not look like APs, rather small increases in fluorescence due to low Ca^2+^ elevations. The authors should also show longer recordings (as they do in Figure 5B for instance) and then compare between the optical responses (DF/F) of jRCAMP to spontaneous APs versus those obtained by the optical stimulation of stCoChR. Are they different (DF/F, kinetics, etc).?

We performed the experiments requested by the referees. Specifically:

1. We performed combined photostimulation and juxtasomal electrophysiological experiments in neurons expressing only jRCaMP1a. Under these experimental conditions, we found that photostimulation did not increase neuronal firing (new Figure 6—figure supplement 1), demonstrating no effect of two-photon holographic stimulation on neuronal firing in the absence of opsin expression. These results are now described at lines 300-301.

2. We performed simultaneous all-optical and juxtasomal electrophysiological recordings in neurons co-expressing jRCaMP1a and stCoChR (new Figure 6F-G). We observed that jRCaMP1a transients induced by photostimulation were associated with the firing of 6.7 ± 1.9 (range: 3-14) action potentials, in agreement with what previously observed in cells expressing only stCoChR (Figure 3B, C). These results are now described at lines 295-299. As requested by the referees, we also compared spontaneous and photostimulation-evoked jRCaMP1a transients (new Figure 6—figure supplement 2). We found that the ∆F/F_0_ of spontaneous events corresponding to the discharge of few action potentials were smaller than the ∆F/F_0_ of photostimulation-evoked jRCaMP1a transients, in agreement with the observation that our photostimulation protocol evoked trains of 3-14 action potentials in stimulated neurons (new Figure 6—figure supplement 2). The decay kinetic of spontaneous and photostimulation-evoked jRCaMP1a transients were not significantly different (new Figure 6—figure supplement 2). These results are now described at lines 301-307.